

# The aerosol-climate model ECHAM6.3-HAM2.3: Aerosol evaluation

Ina Tegen[1], David Neubauer[2], Sylvaine Ferrachat[2], Colombe Siegenthaler-Le Drian[3], Isabelle Bey[3,a], Nick Schutgens[4,b], Philip Stier[4], Duncan Watson-Parris[4], Tanja Stanelle[2], Hauke Schmidt[5], Sebastian Rast[5], Harri Kokkola[6], Martin Schultz[7], Sabine Schroeder[7], Nikos Daskalakis[8], Stefan Barthel[1], Bernd Heinold[1], and Ulrike Lohmann[2]

[1]Leibniz Institute for Tropospheric Research, Leipzig, Germany
[2]Centre for Climate Systems Modeling (C2SM), ETH Zurich, Switzerland
[3]Department of Physics, University of Oxford, United Kingdom
[4]Institute of Atmospheric and Climate Science, ETH Zurich, Switzerland
[5]Max Planck Institute for Meteorology, Hamburg, Germany
[6]Atmospheric Research Centre of Eastern Finland, Finnish Meteorological Institute, Kuopio, Finland
[7]Forschungszentrum Juelich, Juelich, Germany
[8]Laboratory for Modeling and Observation of the Earth System (LAMOS), Institute of Environmental Physics (IUP), University of Bremen, Bremen, Germany
[a]now at: MeteoSwiss, Geneva, Switzerland
[b]now at: Faculty of Life and Earth Sciences, Vrije Universiteit, Amsterdam, The Netherlands

*Correspondence to:* I. Tegen (itegen@tropos.de)

**Abstract.** We introduce and evaluate the aerosol simulations with the global aerosol-climate model ECHAM6.3-HAM2.3, which is the aerosol component of the fully coupled aerosol-chemistry-climate model ECHAM-HAMMOZ. Both the host atmospheric climate model ECHAM6.3 and the aerosol model HAM2.3 were updated from previous versions. The updated version of the HAM aerosol model contains improved parameterizations of aerosol processes such as cloud activation, as

5  well as updated emission fields for anthropogenic aerosol species and modifications in the online computation of sea salt and mineral dust aerosol emissions. Aerosol results from nudged and free running simulations for the 10-year period 2003 to 2012 are compared to various measurements of aerosol properties. While there are regional deviations between model and observations, the model performs well overall in terms of aerosol optical thickness, but may underestimate coarse mode aerosol concentrations to some extent, so that the modeled particles are smaller than indicated by the observations. Sulfate

10  aerosol measurements in the US and Europe are reproduced well by the model, while carbonaceous aerosol species are biased low. Both mineral dust and sea salt aerosol concentrations are improved compared to previous versions of ECHAM-HAM. The evaluation of the simulated aerosol distributions serves as a basis for the suitability of the model for simulating aerosol-climate interactions in a changing climate.



## 1  Introduction

The increase in the positive radiative forcing of anthropogenic greenhouse gases and tropospheric ozone is partly offset by aerosols imposing a negative radiative forcing (Boucher et al., 2013; Myhre et al., 2013). Global aerosol-chemistry-climate models are key tools in the attribution and projection of the role of aerosols in the climate system. In general, aerosol com-
ponents such as black and organic carbon, sulfate, mineral dust and sea salt aerosols are considered in such models as well as their sources, sinks, transport and chemical and microphysical transformations. Considerable efforts have been made over the last decades to improve the incorporation of the relevant aerosol processes in climate models that control the distribution and effects of these species in the atmosphere. However, uncertainties in quantifications of aerosol-radiation interactions and aerosol-cloud interactions remain large. Further development and evaluation of global climate-aerosol-chemistry models is
thus necessary to reduce such uncertainties and provide a basis for investigating the response of the coupled aerosol-climate system in a changing climate.

As well as the host climate models, embedded aerosol-chemistry models are continuously refined and further developed as new processes are included and process representations are improved. The increasing complexity of these models requires systematic documentation of the different existing versions. The ECHAM-HAM model consisting of the atmospheric general
circulation model ECHAM and the aerosol module HAM has previously been widely used in process studies (Lohmann and Hoose, 2009; Folini and Wild, 2011; Kazil et al., 2012; Peters et al., 2014; Neubauer et al., 2014; Schutgens et al., 2014; Gasparini and Lohmann, 2016; Lohmann and Neubauer, 2018) and contributed extensively to model evaluation and intercomparison studies (Textor et al., 2006, 2007; Kulmala et al., 2011; Stier et al., 2013; Jiao et al., 2014). The latest version of the ECHAM-HAMMOZ model (version ECHAM6.3-HAM2.3-MOZ1.0) combines the most recent versions ECHAM (ECHAM6,
Stevens et al. (2013), the aerosol module HAM2 (Zhang et al., 2012) and the atmospheric trace gas chemistry module MOZ (described in Rast et al. (2014)). The aerosol (HAM) and the chemistry (MOZ) modules can be used either interactively or independently of each other. The coupled ECHAM6-HAMMOZ model is described in detail in Schultz et al. (2017). The notation ECHAM-HAMMOZ is used when both the aerosol and chemistry modules are used interactively in combination with the climate model ECHAM, and the notations ECHAM-HAM and ECHAM-MOZ apply when only the aerosol and chemistry mod-
ules, respectively, are used individually. The HAM and MOZ modules share a common interface with ECHAM6 and consistent representation of common processes (e.g., emissions and deposition of trace gases/aerosols as well as cloud microphysics) and the associated routines. The details of the chemistry module MOZ and evaluation of the ECHAM6.3-HAM2.3-MOZ1.0 model configuration is described in Schultz et al. (2017). Cloud processes and cloud-aerosol interactions as well as direct radiative forcing simulated in ECHAM6.3-HAM2.3 are evaluated in Neubauer et al.(submitted).
In this paper the emphasis is placed on the description and evaluation of the aerosol distributions simulated by ECHAM6.3-HAM2.3 to provide a basic quantitative evaluation against a suite of observations of the different aspects of aerosol distributions. We focus on the model version using the modal aerosol computing microphysical processes such as as nucleation, coagulation and condensational growth by the modal scheme M7 (Vignati et al., 2004; Zhang et al., 2012; Neubauer et al.,

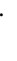 

2014; Schutgens et al., 2014). Alternatively the aerosol microphysical processes can be described by the sectional aerosol scheme SALSA in the ECHAM6.3-HAM2.3-SALSA configuration, which is described in Kokkola et al. (2008, 2018).

## 2  Model description

### 2.1  Model development overview

The aerosol module HAM was first implemented in the 5th generation of the atmospheric general circulation model ECHAM (ECHAM5, Roeckner et al. (2003)) by Stier et al. (2005). In the past years, ECHAM-HAM has undergone substantial software restructuring and scientific development. The host atmospheric model ECHAM was considerably further developed and improved, leading to the version ECHAM6 (Stevens et al., 2013). The HAM module has been continuously expanded with new processes based on the version HAM2 as described in Zhang et al. (2012). The MOZ module for tropospheric and strato-

spheric chemistry was subsequently introduced in a joint effort by several institutions. The first version of the fully coupled aerosol-chemistry-climate model ECHAM5-HAMMOZ was documented in Pozzoli et al. (2008). The latest version of the ECHAM-HAMMOZ model has been developed as an international collaboration. The model is currently hosted by ETH Zurich (Switzerland) and TROPOS in Leipzig (Germany) (https://redmine.hammoz.ethz.ch/projects/hammoz).

The recent generation of the ECHAM-HAMMOZ model is constructed in a more modular approach compared to previous

versions to minimize interactions of the aerosol module with the host general circulation model. ECHAM6 now provides a generic sub-model interface, i.e. a specific FORTRAN module, which contains all calls to the aerosol and chemistry routines. This facilitates simultaneous development and separation of the climate (ECHAM), chemistry (MOZ), and aerosol (HAM) modules. The structure of the aerosol and gas-phase chemistry codes was harmonized so that both components use the same routines for emissions, dry deposition and washout (with adaptations as necessary due to the differences in the respective

processes). The tracer interface for the definition of chemical species including their physical and chemical properties, and the concept of output streams to allow for flexible output of tracer diagnostics including tracer mass mixing ratios, emission, dry deposition, and washout mass fluxes for selected tracers was further extended. It allows for example to distinguish between species that define the physical and chemical aerosol properties, and tracers that essentially provide the memory for advected compounds. While for gas-phase compounds species and tracers are identical, individual aerosol species can be contained in

several tracers such as different aerosol modes or size bins.

### 2.2  ECHAM6

ECHAM is an atmospheric general circulation model developed by the Max Planck Institute for Meteorology in Hamburg, Germany. The model utilizes a spectral transform dynamical core and a semi-Lagrangian tracer transport scheme in flux form (Lin and Rood, 1996). Vertical transport considers turbulent mixing, moist convection (shallow, deep, and mid-level

convection), and momentum transport by gravity waves. Convection is parameterized via the mass-flux schemes by Tiedtke





(1989) and Nordeng (1994). Parameterization of sub-grid scale stratiform clouds uses the scheme of Sundqvist et al. (1989). Cloud liquid water content and cloud ice mixing ratios are computed prognostically (Lohmann and Roeckner, 1996).

The current version ECHAM6 is described in detail in Stevens et al. (2013). The vertical discretization within the troposphere (in particular in the upper troposphere and lower stratosphere) is slightly different in ECHAM6 compared to the previous version ECHAM5. The representation of convective triggering has been improved, and the tuning of various model parameters was adjusted. ECHAM6 is frequently used in a middle-atmosphere configuration with the two verticals grids L47 and L95 that resolve the atmosphere from the surface up to 0.01 hPa (roughly 80 km). Radiative transfer in ECHAM6 is computed using the PSrad/RRTMG (Rapid Radiative Transfer Model for GCMs) (Iacono et al., 2008; Pincus and Stevens, 2013) radiation package, which considers 16 bands for the shortwave (820 to 50000 cm$^{-1}$) and 14 bands for the longwave (10 to 3000 cm$^{-1}$) parts of the spectrum, respectively. Optical properties of clouds are pre-calculated for each band of the RRTMG scheme using Mie theory and read from look-up tables. The concentration of cloud droplet number concentrations are prescribed differently over land and ocean in case ECHAM is used without the HAM aerosol module. In this case climatological average aerosol optical properties by Kinne et al. (2013) are used in radiative transfer computations in ECHAM6. Trace gas concentrations of long-lived greenhouse gases are specified in the model if used without chemistry module. ECHAM6 includes the land-surface model JSBACH (Reick et al., 2013) that assumes that each land grid cell is composed of two fractions, representing bare and vegetated soil surfaces. The vegetated surface fraction is further sub-divided into tiles for each of the plant functional types distinguished in JSBACH. Soil hydrology is represented with a single-layer bucket model.

The variability in the tropics continues to be well represented in ECHAM6 similarly to its predecessor ECHAM5 (Roeckner et al., 2003). This includes e.g. intraseasonal variability, the quasi-biennial oscillation, and some aspects of the El Niño Southern Oscillation (ENSO). The representation of extratropical circulation is clearly improved in ECHAM6 (Stevens et al., 2013).

Compared to the original version of ECHAM6 the updates in the current version ECHAM6.3 include some modifications in the radiation and land surface schemes, and an improved submodel interface. The influence of orography on surface roughness was replaced by a aerodynamic roughness determined by vegetation cover.

ECHAM drives the aerosol and chemistry modules through the generic sub-model interface by providing meteorological conditions such as wind, temperature, pressure, humidity and conditions related to the land surface (taken from JSBACH) such as Leaf Area Index (LAI). Aerosols and their precursors are transported analogous to the tracer transport of water vapor and cloud water in ECHAM.

### 2.3 HAM2

The Hamburg Aerosol Model HAM (Stier et al., 2005) computes the evolution of an aerosol mixture considering the species sulfate, black carbon (BC), organic carbon (OC), sea salt, and mineral dust. Coupled to an atmospheric general circulation model such as ECHAM, the development of mass and number concentrations of the aerosols is computed taking into account physical and chemical particle processes. In turn, the effects of aerosols on clouds and radiation are computed prognostically in the coupled ECHAM-HAM model. The second model version HAM2 containing new updates in parameterizations of particle nucleation and growth, emission calculations for natural aerosol species and aerosol-cloud interactions is described




and evaluated by Zhang et al. (2012). The relative importance of the individual aerosol processes in ECHAM5-HAM2 have been evaluated by Schutgens et al. (2014).

The default version of HAM describes the aerosol size spectrum by the modal M7 aerosol model (Vignati et al., 2004). Aerosols are simulated as superposition of seven log-normal modes: Nucleation mode, soluble (mixed) and insoluble Aitken,

accumulation and coarse modes (Table 1). The aerosol distribution in each mode is described by the aerosol number, the median radius, and the standard deviation. The standard deviation is 1.59 for the nucleation, Aitken, and accumulation modes and 2.00 for the coarse modes. The median radius of each mode is calculated from the aerosol number and aerosol mass, which are transported as tracers within the respective mode. Each aerosol mode is assumed to be internally mixed such that individual particles in a mode can consist of different species. To be considered soluble, at least one species within a particle must be

soluble. Insoluble particles can become mixed (soluble) through condensation of soluble substances. and collisions with mixed particles.

**Table 1.** Aerosol size modes and species in the M7 aerosol microphysics in HAM. Mode boundaries for the is the number median particle radii $\bar{R}$ are given for each mode

| Size mode | Soluble | Insoluble |
|---|---|---|
| Nucleation ($\bar{R} < 0.005\mu$m) | Sulfate | |
| Aitken ($0.005\mu$m$< \bar{R} < 0.05\mu$m) | Sulfate, OC, BC | OC, BC |
| Accumulation ($0.05\mu$m$< \bar{R} < 0.5\mu$m) | Sulfate, OC, BC, sea salt dust | Dust |
| Coarse ($\bar{R} > 0.5\mu$m) | Sulfate, OC, BC, sea salt, dust | Dust |

The current version HAM2.3 described here is updated in terms of default settings and model organization, aerosol emissions, water uptake, wet deposition, and aspects of aerosol-cloud interactions compared to the version HAM2.0 described by Zhang et al. (2012). In addition to minor corrections and bugfixes, major changes in HAM2.3 are:

• Updates and changes in emissions of aerosols and aerosol precursors from anthropogenic and natural sources (described in detail in section 2.3.1):

   – New emission datasets for anthropogenic emissions of BC, OC and $SO_2$

   – Updated emission parameterization for mineral dust

   – New emission parameterization for sea salt aerosols based on Long et al. (2011) and Sofiev et al. (2011) including

parameterization for temperature dependence

   • Modified aerosol-cloud interactions (described in Lohmann and Neubauer (2018)):

   – Cloud droplet activation according to Abdul-Razzak and Ghan (2000) based on Köhler theory

   – Updated treatment of cloud droplet number concentrations (CDNC) detrained from convective clouds

   – Size-dependent in-cloud scavenging by Croft et al. (2010)





- Assuming hexagonal plates as shape of ice crystals following Pruppacher and Klett (1997)

- Limiting immersion freezing of black carbon to particles in the accumulation or coarse mode

- Changed temperature dependence of sticking efficiency for accretion of ice crystals by snow according to Seifert and Beheng (2006)

- Optional choice of minimum CDNC as either 40 cm$^{-3}$ or 10 cm$^{-3}$

### 2.3.1 Emissions of aerosol particles and aerosol precursors

The HAM2.3 emission module of primary aerosol particles and gas-phase compounds has been designed such that emissions are specified for individual sectors in a user-friendly way. An emission input file specifies for each species which emission sectors are considered and how the emission fluxes from these sources are introduced in the model simulation. For example,

all species can be emitted into the lowest model level, or a model level corresponding to a specific altitude, or emitted species can be evenly mixed within the planetary boundary layer. This applies to all emissions from a specific sector. It is also easily possible to apply a scale factor to emission fluxes from a specific sector. This factor can also be used to temporarily turn off individual emission types or sectors.

The default version of ECHAM6.3-HAM2.3 uses the Atmospheric Chemistry and Climate Model Intercomparison Project

(ACCMIP) emission dataset (Lamarque et al., 2010) for anthropogenic and biomass burning emissions. It is based on horizontally gridded temporally interpolated monthly mean emissions for the years 1850 to 2000 of anthropogenic emissions combined from regional and global inventories, and is available at 0.5° horizontal grid resolution. SO$_2$, BC, and OC emissions are considered for the relevant anthropogenic sectors including agricultural waste burning, aircraft, domestic, energy, industry, ships, transport and waste. The dataset also contains biomass burning emission fields with historical emissions. These were

available at decadal increments and were further interpolated at yearly resolution (see http://aerocom.met.no/emissions.html for details) and degraded to the T63 resolution. From 2000 to 2100 this dataset is created from linear time interpolation of the future emission projections. They can be chosen from four different Representative Concentrations Pathways (RCPs), RCP2.6, RCP4.5 RCP6 and RCP8.5 (van Vuuren et al., 2011), denoting the radiative forcing target levels for the year 2100 of 2.6, 4.5, 6 and 8.5 Wm$^{-2}$, respectively. The interpolated anthropogenic ACCMIP and RCP8.5 emis-

sions for the years 1850 and 1960 to 2010 are identical to the AeroCom-II ACCMIP hindcast emission sources available at http://aerocom.met.no/download/emissions/AEROCOM-II-ACCMIP/. The biomass burning emissions for forest and grass fires in this emission dataset represent average conditions of the respective decade. Interannual variability of biomass burning is not considered. Injection heights of biomass burning emissions follow the recommendations of Val Martin et al. (2010). 75% of the emissions are evenly distributed within the planetary boundary layer (PBL), 17% in the first level and 8% in the second

level above the PBL.

In addition to ACCMIP, other datasets can be used to prescribe species emissions. For biomass burning, the Global Fire Assimilation System (GFAS) (Kaiser et al., 2012) can be used alternatively. GFAS provides gridded biomass burning emissions at 0.5° horizontal grid resolution assimilated from fire radiative power from MODIS satellite observations. For ECHAM6-HAM2.3 the fire emissions for BC, OC and SO$_2$ and dimethyl sulfide (DMS) are used from this emission dataset. Combustion





rates are computed using conversion factors for specific land covers. Kaiser et al. (2012) recommend to scale the particulate emissions from the GFAS emission files by the factor 3.4 in order to optimally match observed aerosol optical thickness. This scaling has been shown to perform well for ECHAM-HAM by Veira et al. (2015). For the evaluation of ECHAM6-HAM2.3 simulations presented in this paper we performed simulations with this scaling factor.

In the HAMMOZ configuration, the secondary volatile organic carbon emissions serving as precursors for secondary organic aerosol (SOA) formation are calculated with an implementation of the MEGAN2.1 model (Guenther et al., 2012; Henrot et al., 2017). SOA formation can be computed with the implementation by O'Donnell et al. (2011) which considers the chemical conversion of volatile organic gases into condensable gases, and the partitioning of semi-volatile condensable species into their gas and aerosol phases. The explicit secondary organic aerosol formation routine is not used in the standard setup of
ECHAM6.3-HAM2.3. Instead biogenic emissions are treated as primary OC emissions following AeroCom (Dentener et al., 2006).

Mineral dust emissions are computed on-line using the dust source scheme of Tegen et al. (2002) with modifications as described in Cheng et al. (2008) and Heinold et al. (2016). Dust particle emissions are driven by the 10-m wind speed computed by the atmospheric model. Emission fluxes follow a nonlinear physical process, which depends on surface features and
meteorological conditions in potential source areas. HAM prescribes a constant low roughness length of 0.001 cm for the dust emission calculations in potential source areas. The explicit formulation of the saltation process follows Marticorena and Bergametti (1995). A ratio between vertical and horizontal emission fluxes is prescribed for each soil type (Tegen et al., 2002). Dust emissions can only take place in potential dust source areas (usually non- or low-vegetated areas), the distributions of which is taken from an external file derived by Tegen et al. (2002), who identified potential dust source areas using the satellite-
derived fraction of vegetated areas and a model-derived distribution of potential vegetation types. ECHAM6.3-HAM2.3 also includes the option of deriving potential dust sources using the vegetation cover provided by the land component JSBACH, which allows a full coupling with the land surface scheme (Stanelle et al., 2014). For Saharan dust sources a satellite-based source mask is implemented (Heinold et al., 2016). It is based on the infrared dust index from the SEVIRI instrument on the geostationary Meteosat Second Generation satellite that allows for identification of realistic spatiotemporal distributions of
dust emission events (Schepanski et al., 2009).

In previous versions, a global correction factor of 0.86 was applied on the threshold friction velocity to account for the inhomogeneity of the factors influencing dust emissions (e.g., surface wind) across the rather coarse model grid boxes. In ECHAM6.3 the surface orography is not taken into account for the aerodynamic surface roughness, in contrast to earlier versions. The subsequent changes in surface wind distributions over dust source areas require additional regional correction
factors to get dust emissions in each relevant region to agree with the previous version. These regional correction factors are set as 1.45 for North- and South America and Asia, and 1.05 for all other regions.

Several parameterizations can be chosen in ECHAM6.3-HAM2.3 for sea salt aerosol emissions. In earlier versions of HAM the parameterization by Guelle et al. (2001) was used in the default setup. In the past years several new sea salt emission parameterizations were developed by different authors mostly based on laboratory measurements. Such measurements also
revealed that the sea salt aerosol emissions depend to a certain extent on the temperature of the surface water, such that at colder





temperatures emissions are lower and led to emission of smaller particles compared to warmer temperatures (e.g., Sofiev et al. (2011)). The new standard in ECHAM6.3-HAM2.3 for sea salt emissions uses a parameterization following Long et al. (2011) taking into account the temperature dependence according to Sofiev et al. (2011). The performances of the different sea salt emission schemes will be compared in section 5.7. The sea salt emissions now use surface wind speed as well as sea surface

temperatures from the model to compute sea salt aerosol emissions for the mixed accumulation and coarse modes. As a marine source for aerosol precursors, natural emissions of dimethyl sulfide (DMS) from the marine biosphere are calculated online following Kloster et al. (2006).

### 2.3.2 Aerosol microphysics

Aerosol processes in M7 (Vignati et al., 2004) include nucleation of sulfuric acid-water droplets, coagulation, condensation

of sulfuric acid and aerosol water uptake. These processes lead to a redistribution of particle numbers and mass among the different modes. For nucleation, the standard version of the model uses the scheme implemented by Kazil et al. (2010), with optional $H_2SO_4$-organic nucleation based on kinetic nucleation theory (Kuang et al., 2008) or cluster activation (Kulmala et al., 2006; Kazil et al., 2010). Condensation of sulfuric acid occurs on all pre-existing particles of all sizes. Intra-modal and intermodal coagulation is considered for the soluble modes (expect the mixed coarse mode) and the Aitken insoluble mode.

Condensation and coagulation increase the geometric mean radii of the mixed modes, allowing smaller particles to grow into a larger mode. Also, formation of a mono-layer coating of sulfate on an insoluble particle causes it to be moved to a mixed (soluble) mode. The water content of aerosols in each mode is calculated from their chemical composition and the ambient relative humidity using a semi-empirical water uptake scheme based on the $\kappa$-Köhler theory (Petters and Kreidenweis, 2007) as implemented by O'Donnell et al. (2011).

In the standard released version of ECHAM6.3-HAM2.3, the representation of SOA is based on the assumption that about 15% of natural terpene emissions at the surface form SOA as described in Dentener et al. (2006). They are assumed to condense immediately on existing aerosol particles and to have identical properties to primary organic aerosols (Stier et al., 2005). As an alternative, an interactive module for the formation of SOA is available (O'Donnell et al., 2011). The SOA precursors considered include biogenic compounds and aromatic compounds from anthropogenic activities and biomass burning. In that

scheme, oxidation of biogenic precursors produces two semi-volatile products that can condense on existing organic-containing particles while the oxidation of the aromatic compounds leads to non-volatile products that condense immediately.

### 2.3.3 Sulfur chemistry

The sulfur chemistry in HAM2 is based on Feichter et al. (1996). Prognostic variables include concentrations of DMS, $SO_2$ and gas- and aqueous-phase sulfate. With the HAM setup (without MOZ), an eight-year mean reanalysis of the atmospheric

oxidants covering the period $2003 - 2010$ is used. This climatology was constructed by assimilating satellite data into a global model and data assimilation system (Inness et al., 2013). Averaged monthly mean oxidant fields include hydroxyl radical (OH), hydrogen peroxide ($H_2O_2$), nitrogen dioxide ($NO_2$), ozone ($O_3$), and nitrate radical ($NO_3$). Sulfuric acid produced from gas-phase chemistry can nucleate to form new particles or condensate on existing aerosol particles. Sulfate produced from aqueous





phase chemistry is distributed to pre-existing particles in the soluble accumulation and coarse modes. For the HAMMOZ setup the sulfur oxidants are computed online taking into account the full atmospheric chemistry processes described by MOZ, see Schultz et al. (2017).

### 2.3.4 Removal processes

Aerosol particles are removed by sedimentation, dry and wet deposition. Gravitational sedimentation of particles in HAM2 is calculated based on their median size using the Stokes settling velocity (Seinfeld and Pandis, 1998), applying a correction factor according to Slinn and Slinn (1980). Removal of aerosol particles from the lowest model layer by turbulence depends on the characteristics of the underlying surface (Zhang et al., 2012). The aerosol dry deposition flux is computed as the product of tracer concentration, air density and deposition velocity, depending on the aerodynamic and surface resistances for each

surface type considered by ECHAM6.3, and subsequently added up for the fractional surface areas. For wet deposition the in-cloud scavenging scheme from Croft et al. (2010) dependent on the wet particle size is used. The in-cloud scavenging scheme takes into account scavenging by droplet activation and impaction scavenging in different cloud types, distinguishing between stratiform and convective clouds and warm, cold, and mixed-phase clouds. Below clouds particles are scavenged by rain and snow using a size-dependent below-cloud scavenging scheme (Croft et al., 2009).

### 15  2.3.5  Aerosol optical properties

Aerosol optical properties properties are dynamically computed when using the prognostic aerosol module in ECHAM6.3-HAM2.3. The effective refractive index of each aerosol mode is computed from volume-weighted averages of the refractive indices and Mie-scattering size parameters of the individual components including the water content, assuming internal mixing (Stier et al., 2007; Zhang et al., 2012). For absorbing aerosol species, the complex refractive index for BC at 550 nm is

$1.8 + 0.71i$ (Bond and Bergstrom, 2006; Stier et al., 2007), and $1.52 + 0.0011i$ for dust aerosol (Kinne et al., 2013). Extinction cross sections, single scattering albedos (SSA) and asymmetry parameters are provided via a look-up table and then re-mapped onto the bands of the ECHAM radiative transfer model.

### 2.4  Cloud microphysics

A detailed description of the current implementation of cloud processes and aerosol-cloud interaction is given in Lohmann

and Neubauer (2018) and Neubauer et al. (submitted). The two-moment cloud microphysics scheme in ECHAM simulating the number concentrations and mass mixing ratios of cloud droplets and ice crystals is coupled to the aerosol scheme HAM through the processes of cloud droplet activation and ice crystal nucleation (Lohmann et al., 2007) as well as through in-cloud and below-cloud scavenging. Processes such as phase changes, growth by water vapor condensation, deposition and collision processes and precipitiation formation are considered (Zhang et al., 2012). Updates in the cloud scheme in ECHAM6.3-

HAM2.3 compared to previous versions include the computation of cloud droplet activation according to Abdul-Razzak and Ghan (2000) based on Köhler theory, limiting immersion freezing of black carbon to particles in the accumulation or coarse





mode, a temperature dependence of sticking efficiency for accretion of ice crystals by snow following Seifert and Beheng (2006) and an option to choose minimum CDNC as either 40 $cm^{-1}$ or 10 $cm^{-1}$. Also, inconsistencies were removed e.g. in the calculation of condensation and cloud cover, as well as in the calculation of the ice crystal number concentration in cirrus clouds. The two-moment cloud microphysics is energy-conserving and has been modularized in the updated version.

## 3 Model set-up and experiments

In this publication, we evaluate different aspects of the simulated aerosol distributions for several simulations from the ECHAM6.3-HAM2.3 model. All simulations were performed in T63 spectral resolution which corresponds to $1.875° \times 1.875°$ horizontal resolution. The vertical resolution is 47 vertical layers with a top at 0.1 hPa. The increased vertical resolution, which affects mostly the stratosphere, has only a limited influence on the global tropospheric aerosol distributions compared to the 31 layers used in the previous version (Zhang et al., 2012; Neubauer et al., 2014). It is used here to ensure consistency with the host model ECHAM. Sea surface temperatures were fixed in the model simulations. The model simulations in this work do not utilize the MOZ submodel or the SOA scheme.

In the base model setup ('NUDGE'), direct comparisons with aerosol observations available at specific dates are facilitated by simulations in a nudged mode, in which vorticity, divergence, and pressure are relaxed towards the ERA-Interim re-analysis (Berrisford et al., 2011). Sea surface temperatures (SSTs) for this model setup were set to AMIP SSTs for the respective year (Taylor et al., 2000). Since the nudging may have some impact on the computation of the aerosol processes, and as the model will be used in a free mode without nudging in most upcoming studies, the results will be compared for a free, not-nudged simulation (labeled 'CLIM'). The standard model setup includes anthropogenic and biomass burning emissions from the ACCMIP dataset, as described in section 2.3.1 with emission projections based on the RCP4.5 scenario. For the time period 2003-2012 considered in this work, the ACCMIP biomass burning emissions are based on scenarios rather than observations, and thus do not vary on daily or interannual timescales. For comparison, aerosol distributions are also simulated with daily-available GFAS biomass burning emissions that are based on satellite retrievals. As described in section 2.3.1 and suggested by Kaiser et al. (2012) the particulate GFAS emissions for biomass burning are multiplied by a factor of 3.4 in simulation GFAS. For the evaluation of the new sea salt emission scheme further sensitivity studies are presented, which are described in section 5.7.

The simulations were carried out for the years 2003 to 2012. This time period overlaps with the new reference period as agreed upon in the AEROCOM project, which is 2003-2010, and with the previous reference period for the ECHAM5-HAM2 simulations that was 2000-2009. For those observations which are time resolved for years within the simulation period, the comparisons are carried out for the actual dates of the observations. Otherwise the evaluation is for the averaged aerosol properties over the simulation time period.



**Table 2.** Set-up of the simulations with ECHAM6.3-HAM2.3.

| Simulation | Description |
| --- | --- |
| CLIM | T63;47 vertical layers; ACCMIP interpolated emissions for $SO_2$, OC, BC; climatological SST |
| NUDGE | As CLIM; nudged meteorology; AMIP SST |
| GFAS | as NUDGE, using GFAS biomass burning emissions multiplied by factor 3.4 |

## 4    Observations

### 4.1    Aerosol optical thickness and Ångstrom exponent

Ground-based information on column aerosol properties are available from the global sunphotometer network AErosol RObotic NETwork (AERONET, http://aeronet.gsfc.nasa.gov/, Holben et al. (1998). Quality-controlled measurements are routinely taken
at several wavelenghts, providing information on aerosol optical depth and and Ångstrom exponents (AE), which are an indication for average effective particle sizes in the atmospheric column. These data are widely used as 'ground truth' for aerosol properties, e.g. for evaluation of aerosol model results and satellite retrievals. Model results are compared to level 2.0 cloud-screened, 6 hour averages of AOT measurements at 675 nm wavelength by linearly interpolating model values to the times and locations of the measurements at the locations of the respective AERONET stations (see Fig. 1). The retrieved AEs derived
from the extinction measurements at 440 and 870 nm wavelengths are compared to collocated modeled values that are computed from simulated AOTs at 550 and 865 nm. Single scattering albedos (SSA) are taken from the L2 AERONET Inversion product (Dubovik and King, 2000; Holben et al., 2006).

The global distribution of modeled AOT is additionally compared with retrievals from the MODerate-resolution Imaging Spectroradiometer (MODIS) instrument on the Aqua satellite (King et al., 1999). We used a data-assimilation grade product
based on Dark Target retrievals, developed by NRL (Naval Research Laboratory) (Zhang and Reid, 2006; Hyer et al., 2011; Shi et al., 2011). For a direct comparison of model results and satellite retrievals the model AOTs were linearly interpolated to the time and the location of the available satellite observations (Schutgens et al., 2017).

### 4.2    Aerosol particle size

Aerosol size distributions were compared with in-situ measurements from several stations described by Asmi et al. (2011) for
the year 2009, and with compiled number size distributions for the Aitken and accumulation modes compiled for different marine regions by Heintzenberg et al. (2000). For the European Supersites for Atmospheric Aerosol Research (EUSAAR; http://www.eusaar.net/ ) particle number concentrations and size distributions in the size range between 30 and 500 nm dry diameter are available for total 24 stations. Here comparisons are done for 15 stations in different European regions (Fig. 2). The observations of number concentrations at the individual sites are converted into lognormal distributions, which facilitates
comparisons of size distributions from the model that are computed as lognormal modes. Heintzenberg et al. (2000) compiled observations from 30 years of marine aerosol measurements and made them available on a $15° \times 15°$ grid that is well suited



for comparisons with global aerosol models. Measured number size distributions for the Aitken and accumulation modes are available. Since these observations were taken before the simulation period, they are used to evaluate the climatological median of the modeled size distribution.

### 4.3 In-situ surface observations of aerosol species concentration

To evaluate the simulated aerosol mass mixing ratios at the surface, we compared the simulated data against those measured by the European Monitoring and Evaluation program (EMEP; http://www.emep.int) and the United States Interagency Monitoring of Protected Visual Environment (IMPROVE; http://vista.cira.colostate.edu/improve/). Both of these observation networks provide data for the mass concentrations of individual chemical components. From the EMEP and IMPROVE monitoring sites we compare the PM10 aerosol mass concentration measurements for sulfate and black carbon. Additionally, for IMPROVE

sites we compare organic carbon concentrations. In total, data from 530 stations are available for the EMEP and the IMPROVE networks, see Fig. 2. Comparisons of surface concentrations of BC and sulfate and EMEP observations were done for the period 2003-2010. Surface mass concentrations of OC were compared against IMPROVE observations for 2003 and 2004. For comparison the simulated concentrations at the model layer that corresponds to the altitude of the station of the compared species were sampled for the days when observations were available at each station and averaged in the same way as the

observations. Moreover, the simulated concentration are collocated to the locations of the individual stations.

Surface mass concentrations of mineral dust and sea salt aerosols were obtained from AtmosphERre-Ocean Chemistry Experiment AEROCE (Arimoto et al., 1995) and the SEa/AiR EXchange program SEAREX (Prospero et al., 1989). Monthly surface mass concentrations are available for 29 sites that are used to evaluate modeled dust and sea salt concentrations. These observations have been extensively used for evaluating dust model results, see e.g., Huneeus et al. (2011). The observation

period for these stations was earlier than the simulation period, so we compare the 10-year average of monthly mean concentrations for the years 2003 to 2012.

### 4.4 Aircraft campaigns

Vertical profiles of simulated BC, OC and SO$_4$ concentrations are compared to data from multiple aircraft campaigns. In Koch et al. (2009) aircraft campaign data for BC are compiled, which provide BC mass concentrations measured by Single Particle

Soot Photometers. Mass concentrations of sulfate and OC measured e.g. by aerosol mass spectrometry or filter measurements were compiled by Heald et al. (2011). The locations of the campaigns are shown in Fig. 2. Compared are the model concentrations in the grid cells that are crossed by the flight routes of the aircrafts, for months when the measurements were taken Watson-Parris et al. (submitted).



## 5  Results

### 5.1  Global distribution

For a general overview of the performance of the ECHAM6.3-HAM2.3 aerosol simulation, the simulated global AOT distributions for the CLIM, NUDGE and GFAS experiments are compared with collocated retrievals from the MODIS Aqua satellite instrument for the example year 2007 (Fig. 3). The main features of the simulated AOTs agree overall with the observed patterns. However, while over land the MODIS comparisons point towards lower AOTs in the model results compared to the satellite retrievals, the model AOTs are overestimated over parts of the tropical and southern hemisphere oceans. Typical maximum concentrations downwind the Sahara and the Sahel are caused by dust and biomass burning aerosol. Maximum AOTs in Eastern Asia are resulting from anthropogenic aerosol sources. The shape of the aerosol plume over the Atlantic originating from the African continent is better matched in the NUDGE than in the CLIM results due to the more realistic large-scale wind fields responsible for long-range aerosol transport in the nudged simulation. For the GFAS results the AOT over the biomass burning regions is better matched in South America compared to the NUDGE results in which AOTs are underestimated, but overestimated in the eastern tropical Atlantic. The difference plots between the model results for the NUDGE and GFAS simulations and MODIS AOT highlight that the model overestimates AOT in the tropical and subtropical ocean regions by more than 0.1, particularly for the GFAS results. This could be due to too high marine aerosol, insufficient deposition of aerosol transported from the continents or too high hygroscopic growth in the model. Both simulations show too low AOT in North America compared to the measurements, where AOT is lower by more than 0.1 compared to the observations. This may point to missing aerosol sources in the model such as ammonium nitrate (Croft et al., 2016) or some secondary organic aerosol species in this region, or too low hygroscopic particle growth.

### 5.2  Aerosol optical thicknesses and and Ångstrom exponents and single scattering albedo at AERONET stations

The modeled AOTs and AEs are directly compared with collocated observations by the AERONET sunphotometer stations mapped in Fig. 1 based on daily retrievals. Time series of simulated and observed AOTs (Fig. 4) shown for selected AERONET stations are monthly averages selected for those days when observations were available. These stations where chosen for typical locations in Europe (Ispra (Italy) and Leipzig (Germany)), Asia (Beijing (China) and Gosan (Korea), North America (Cart site and GSFC (USA)), South America (Alta Floresta and Sao Paulo (Brazil) ), Africa (Capo Verde, Banizoumbou (Niger))and Australia (Canberra, Lake Argyle). The magnitudes and temporal variations in AOT for the NUDGE, CLIM and GFAS simulations are mostly well matched with the observations. Seasonal and interannual variabilities are generally well reproduced in the model. The better match of the results from the nudged simulations compared to CLIM in stations largely impacted by long-range transported aerosol such as Capo Verde is evident. At stations where the aerosols from biomass burning contribute significantly to AOT, differences between NUDGE and GFAS results are also clearly evident (Alta Floresta, Sao Paulo). For GFAS the individual AOT maxima and year-to-year differences are better matched with the observations compared to the CLIM and NUDGE results due to the biomass burning emissions based on actual satellite retrievals. In contrast, projected values of the ACCMIP emissions are used in the NUDGE and CLIM experiments. While for the CLIM





simulation the individual AOT maxima are less well matched compared to the NUDGE simulation, the seasonal changes are generally in reasonable agreement with the observations, indicating the important role of the seasonality of emissions and atmospheric processes in addition to the accurate transport patterns. While at most stations the magnitude of the AOTs are well matched between model and observations, there are some exceptions: E.g. at the Ispra site in northern Italy all model results

underestimate the measurements by about a factor 2, and at the station GSFC in Maryland, USA the observed seasonal cycle is not reproduced. The underestimation of AOT in the model at the location of Ispra may be explained by a misrepresentation of the topography at the location near the foothills of the Alps and thus the atmospheric flows. Otherwise, even in highly polluted urban locations such as Beijing the model results and observations are well matched in terms of magnitude and temporal variations at monthly and interannual timescales. The same is the case for locations with very low AOT (Canberra).

In addition, the model results are also provided as scatter plots (Fig. 5). The values are selected for those days when measurements were available and then averaged for the respective year. Almost all annual AOT averages are well within one order of magnitude of the observations. The correlation coefficient for NUDGE and AERONET AOTs is 0.73. The average normalized (by the mean value) root mean square error is 1.2 for the NUDGE results, slightly better than for CLIM with 1.3. On the other hand, the normalized bias is lowest for GFAS results with 0.01. The model results have a slight negative bias of

-0.4 (NUDGE) and -0.3 (CLIM, GFAS). The ratio of standard deviation for model and observations is between 0.84 (NUDGE) and 0.98 (CLIM). That the GFAS simulation compares slightly better to the observations than the NUDGE results reflects the role of the annually varying emissions from biomass fires based on satellite data in GFAS. In particular in the GFAS simulation the agreement is better for North and South America for locations that have annual average AOT values lower than 0.1, where the ACCMIP emission scenario used in the NUDGE experiment leads to too low AOTs in the model.

The simulated Ångstrom exponents (AE) giving an indication of effective aerosol particle sizes in the atmospheric columns are also compared with the AERONET data (Fig. 6). The correlation of the observed and simulated AE of 0.56 for the NUDGE results is lower than the correlation for AOT. It can be expected that modal schemes such as HAM better simulate mass mixing ratios as size distributions of aerosols. Root mean square errors of about 0.3 are similar for all model results. Compared to the observations, the simulated values have a positive bias particularly in North African, South American and oceanic regions,

which means that the simulated particle sizes are too small. The bias in regions that are dominated by dust and sea salt aerosol reflect the fact that natural coarse mode aerosol particles may not be well represented in the modal aerosol scheme. The AE values in the GFAS simulation have a slightly higher bias (0.11) compared to the NUDGE simulation (0.08). The positive AE bias in South America where the aerosol load is strongly impacted by biomass burning aerosols could be an indication that biomass burning aerosols may contain more coarse mode aerosol than assumed in the model. For the AE values at North

American sites (red symbols) the AE values vary more strongly in the model than in the observations in all experiments, which is not the case for the AOTs. Other than possible contributions of secondary organics which may be misrepresented in this model setup, this bias may also be caused by sporadic dust events in this region that are not simulated in the model, but would lead to lower observed AEs at times of dust emissions. However, this would lead to higher dust variability in the observations than in the model, which is not found.





Annual cycles of AOT, AE and SSA are shown for averaged results for the AERONET stations indicated in Fig. 7 and four regions (East Asia, Amazon, Sahara, Southern Ocean) in Fig. 8. AOT model results for NUDGE, GFAS and CLIM are compared to AERONET direct sun retrievals at 675 nm, while SSA from the model is compared to the AERONET inversion product (Holben et al., 2006) at 550 nm. For AOT and AE the AERONET stations used for this comparison were selected

as being regionally representative, as in Kinne et al. (2013). For the timeseries the individually collocated model data and observations were aggregated over regions and 10 days. In the global average the modeled AOT underestimates the observations by values of about 0.05 to 0.1 in the different simulations, with best agreement in northern hemisphere (NH) spring months when AOT is highest. The seasonal AOT pattern is better matched for NUDGE and GFAS than for CLIM model results due to the more realistic transport patterns. The observed NH fall maximum is due to aerosol from the biomass burning smoke

in the Amazon region, which is matched by the GFAS results due to the realistic seasonal distribution of biomass burning emissions in that simulation. The CLIM results underestimate AOT in the Amazon in the NH fall season and the Sahara in all seasons except the winter months. Mineral dust aerosols dominate the aerosol composition in the Sahara region and are produced by strong surface winds. Here, the CLIM results deviate clearly from the results with the nudged model, which could also be seen in the daily results above. Except in East Asia where the aerosol is dominated by anthropogenic aerosol

the AE model results are higher than the observations in agreement with the scatterplot Fig. 6. Again this can be interpreted by the model underestimating the particle size for coarse mode aerosol particles like mineral dust or sea salt. The SSA links the aerosol properties resulting from particle size and composition to their absorption and thus their radiative effect (see also Neubauer et al.). The model results lie slightly below the AERONET inversions in all regions. In the global mean, the retrieved AERONET SSA values vary between 0.88 and 0.95, with values as high as 0.98 in the Sahara, and as low as 0.8 during some

months in the Amazon and East Asia due to high black carbon loads. In some instances the modeled SSA fall below 0.8. The overall slightly lower modeled SSA compared to the AERONET inversions may result in a solar aerosol absorption that is biased high in the model results. On the other hand the too low particle size in coarse mode mineral dust that is indicated by the overestimate of AE in mineral-dust dominated region could result in a too high SSA in the model, which would result in an underestimate of aerosol absorption.

### 5.3 Size distribution

Aerosol size distributions are compared for seasonal averages in the NUDGE simulation to observations at several EUSAAR stations (Asmi et al., 2011) representing different European regions (Fig. 9). Only Aitken and accumulation modes were measured, therefore only these modes are considered in the comparisons. Agreements of number concentrations, particle size distributions and seasonal variations are evident for many of the stations, particularly notable at stations in the northern

and western parts of Europe. In central Europe the number size concentrations are underestimated at the stations K. Puszta and Kosetice, the same is the case for the station Ispra in northern Italy particularly in the winter season. For Ispra this underestimate in number size concentrations is consistent with the underestimated AOTs in this location shown in Fig. 4. As mentioned above, this discrepancy may be due to insufficient resolution of the regional topography and thus too strong mixing of air masses in this region. Also, the model underestimates the maximum number concentration in southern European





stations in summer in Finokalia and Monte Cimione. At other seasons the agreement is better at least at the latter location. At the high altitude stations Puy de Dome and Jungfraujoch some misrepresentations of maximum number size concentrations occur, whereby the concentrations are clearly overestimated in the summer months at Puy de Dome, and the Aitken mode concentrations are overestimated at Jungfraujoch in the model compared to the observations. The same is the case in the high

latitude Zeppelin station. Overall the agreement is good in most cases, considering that global model simulation results are compared to measurements at individual station locations that may not be representative for large areas (Schutgens et al., 2016) .

For larger regions, comparisons of particle number size distributions averaged for oceanic latitudinal bands as compiled by Heintzenberg et al. (2000) (Fig. 10) show generally good agreement in terms of mode sizes and concentration maxima (note

that here the y-axes for the number concentration are logarithmic in contrast to the linear axes used in Fig. 9). Only comparisons for the NUDGE experiment are shown here. The comparisons for the CLIM and GFAS simulations give very similar results in terms of aerosol number size distributions. The shapes of the size distributions and maximum concentrations agree generally well. The widths of the modes of size distributions are slightly larger for the model than the observations in many regions. In the tropics, in particular for the region 0-15° N, the maximum number size concentrations are too low by nearly an order of

magnitude in the model compared to the observations. At northern and southern high latitudes the number size distributions in the model are shifted to smaller sizes compared to the observations. However the distribution at mid latitudes compare well considering that the time period of the observations and the model do not agree.

While the comparison of simulated AE with sunphotometer measurements in Fig. 6 indicates a possible positive bias in the model which hints towards too small particle sizes in the model, this is in general not evident in this direct comparison

of particle size distributions at the surface. However since coarse mode particles were not included in the size distribution measurements, the model's ability to realistically simulate coarse mode particles e.g. for mineral dust and sea salt can not be evaluated with these measurements. Alternatively, hygroscopic particle growth or may be too low in the model.

## 5.4 Aerosol species

The global aerosol species budgets for burdens, emissions, sinks and lifetimes for the CLIM, NUDGE and GFAS experiments

are summarized in Table 3. Here the burdens are also compared with the previous version ECHAM5-HAM2.0 (Zhang et al., 2012) and also with results from the AeroCom aerosol model intercomparion (Textor et al., 2006). All values of the budgets for the individual aerosol species that were computed with the model are within the range of the AeroCom values. While the values did not considerably change compared to the earlier version by Zhang et al. (2012) for the mostly anthropogenic species $SO_4$, BC and OC, differences for dust and sea salt emissions are evident. Dust emissions increased from about 900

Mt/year to 1100 Mt/year due to the regional tuning and are thus closer to the AeroCom average of 1800 Mt/year. However, the magnitude of dust mass emission fluxes also depends on the size range considered in the dust emission calculation. Particle sizes exceeding several micrometers can cause high emission fluxes but do not considerably contribute to atmospheric burdens due to their fast sedimentation rates. Due to slightly increased atmospheric lifetimes in the current model version, global and annually averaged dust burdens increased from 11 to about 17 Tg, also in agreement with the AeroCom average burden of 19.2



Tg. Sea salt mass emissions were considerably reduced by more than a factor of four with the new emission parameterization compared to the earlier version, and as a consequence deposition fluxes and atmospheric burdens of sea salt aerosol were also reduced. The atmospheric sea salt burden is reduced by a factor of about 2-3, which is less than the reduction in emissions. This is consistent the nearly doubled atmospheric lifetimes of the sea salt particles compared to the earlier model version, which

is a consequence of the smaller particle sizes in the new parameterization, ignoring the super-coarse sea salt fraction which deposits very quickly.

### 5.5 Comparison of Sulfate, OC, BC with observations

The locations of the EMEP and IMPROVE stations as well as the flight patterns of the research flights used for comparisons of model results and measurements for the species $SO_4$, OC and BC are shown in Fig. 2.

### 5.5.1 Sulfate

The comparison of sulfate aerosols with surface concentration measurements at EMEP and IMPROVE stations (Fig. 11) shows that the different simulations agree similarly well to the observations for the three experiments. The statistical values given in each panel are root mean square error RMS: (in brackets: normalized RMS); absolute bias (normalized bias); R: Correlation coefficient (R on log scale) and sigma: ratio between simulated and observed standard deviation. For all experiments the

correlation coefficients between modeled and measured surface concentrations are 0.84-0.85 for the comparison at EMEP and the IMPROVE stations, showing that simulated surface concentrations of sulfate aerosol are not affected by different biomass burning emissions in these locations. Also, for the secondary sulfate particles the use of nudged meteorology does not significantly improve the distribution of the simulated particles compared to the free simulation CLIM. The biases of the averaged model results compared to the observations are low.

The comparison to aircraft measurements (Fig. 12) are mostly within the error bars for the observations in the figure that indicate the measurement variabilities. In particular reasonable agreement is found in the free troposphere within the different experiments and comparisons with observations. In the Sahel regions the results for the AMMA campaign show 4-5-fold overestimates in sulfate concentrations at heights between 2 and 4 km compared to the measurements, which may be related to low dry deposition velocities of $SO_2$ over bare soils. While the NUDGE and GFAS results are mostly in close agreement

as the emissions of the sulfate precursor $SO_2$ from biomass burning are generally low compared to anthropogenic emissions, the results from the CLIM simulations deviate considerably from the other results e.g. for the AMMA and OP3 campaigns, indicating that for vertical distribution the use of realistic wind speeds and directions to simulate aerosol transport is important when evaluating $SO_4$ concentrations with aircraft measurements.

### 5.5.2 Black carbon

As for sulfate, the simulated BC aerosol concentrations are compared to in-situ measurements by the EMEP and IMPROVE measurements in Europe and North America (Fig. 13). There is a negative bias in the model simulation compared to the



**Table 3.** Aerosol global annual budgets averaged over 2003-2012. Compared are the results from ECHAM6.3-HAM2.3 with the earlier version ECHAM5-HAM2.0 as described in Zhang et al. (2012) (table 8 therein) and the simulations CLIM, NUDGE and GFAS for the years 2003-20102. The results are also compared with the multi-model AeroCom results by Textor et al. (2006). *For the AeroCom results the dry deposition also contains the sedimentation fluxes.

| | ECHAM5-HAM2.0 | ECHAM6.3-HAM2.3 | | | AeroCom |
| | Zhang et al. 2012 | CLIM | NUDGE | GFAS | Textor et al. 2006 |
|---|---|---|---|---|---|
| **Sulfate** | | | | | |
| Burden (Tg S) | 0.85 | 0.74 | 0.78 | 0.81 | 0.67 |
| Sources (Tg S yr$^{-1}$) | | | | | |
| Emissions + Production | 70.9 | 73 | 73 | 74 | 59.7 |
| Sinks (Tg S yr$^{-1}$) | | | | | |
| Sedimentation | 1.56 | 0.70 | 0.72 | 0.71 | |
| Dry deposition | 2.33 | 2.08 | 2.11 | 2.15 | 6.9* |
| Wet deposition | 66.6 | 69.9 | 69.4 | 71. | 53. |
| Lifetime (days) | 4.4 | 3.7 | 4.0 | 4.0 | 4.1 |
| **BC** | | | | | |
| Burden (Tg) | 0.13 | 0.14 | 0.14 | 0.26 | 0.24 |
| Sources (Tg yr$^{-1}$) | | | | | |
| Emissions | 7.7 | 8.1 | 8.1 | 12.5 | 11.9 |
| Sinks (Tg yr$^{-1}$) | | | | | |
| Sedimentation | 0.02 | 0.02 | 0.02 | 0.03 | |
| Dry deposition | 0.64 | 0.71 | 0.74 | 0.93 | 2.6* |
| Wet deposition | 7.1 | 7.4 | 7.5 | 11.6 | 9.4 |
| Lifetime (days) | 5.9 | 6.3 | 6.5 | 7.5 | 7.1 |
| **OC** | | | | | |
| Burden (Tg) | 1.5 | 1.0 | 1.0 | 2.2 | 1.7 |
| Sources (Tg yr$^{-1}$) | | | | | |
| Emissions | 68 | 69. | 69. | 123. | 97. |
| Sinks (Tg yr$^{-1}$) | | | | | |
| Sedimentation | 0.19 | 0.18 | 0.19 | 0.32 | |
| Dry deposition | 4.5 | 5.4 | 5.6 | 7.5 | 19.2* |
| Wet deposition | 60.3 | 64.4 | 64.4 | 116. | 76.7 |
| Lifetime (days) | 8.4 | 5.4 | 5.5 | 6.6 | 6.5 |
| **Dust** | | | | | |
| Burden (Tg) | 11.6 | 16.5 | 17.9 | 17.3 | 19.2 |
| Sources (Tg yr$^{-1}$) | | | | | |
| Emissions | 805 | 1124 | 1145 | 1107 | 1840 |
| Sinks (Tg yr$^{-1}$) | | | | | |
| Sedimentation | 341 | 370 | 387 | 378 | |
| Dry deposition | 56. | 77. | 70. | 68. | 1235* |
| Wet deposition | 410 | 687 | 696 | 669 | 607 |
| Lifetime (days) | 5.4 | 5.3 | 5.7 | 5.7 | 4.1 |
| **Sea salt** | | | | | |
| Burden (Tg) | 11.6 | 3.9 | 3.9 | 3.9 | 6.4 |
| Sources (Tg yr$^{-1}$) | | | | | |
| Emissions | 6110 | 1212 | 1101 | 1092 | 6280 |
| Sinks (Tg yr$^{-1}$) | | | | | |
| Sedimentation | 2038 | 255 | 244 | 243 | |
| Dry deposition | 1484 | 98 | 82 | 81 | 4377* |
| Wet deposition | 2591 | 863 | 778 | 770 | 1902 |
| Lifetime (days) | 0.69 | 1.2 | 1.3 | 1.3 | 0.41 |





observations, which is reduced in the GFAS experiment. The correlations (R-values between 0.54 and 0.57) are lower than for sulfate. Particularly for concentrations lower than 0.5 $\mu$g m$^{-3}$ the model underestimates the observed surface concentrations, which may be caused by too low local emissions or too fast removal of the particles.

The comparisons to aircraft data for BC use the same observations as the BC AeroCom model intercomparison study by

Koch et al. (2009). For flights in low and mid latitudes (AVE Houston, CR-AVE, TC4, CARB) the model overestimates the BC concentrations in the free troposphere in most cases, which may be due to either too strong vertical transport or too low removal above the boundary layer. Similar overestimates were found for most models compared by Koch et al. (2009). For the flights at high latitudes (ARCTAS, ARCPAC) the GFAS simulations agree well with the observations while the CLIM and NUDGE results underestimate BC concentrations in the boundary layer. Above 200 hPa altitude the modeled BC concentrations remain

quite constant for all simulations. Since in the compared aircraft studies no measurements were taken at those high altitudes it is not clear if the modeled BC distribution at high altitudes is realistic.

### 5.5.3 Organic carbon

The comparisons of OC concentrations with in-situ measurements is similar to the evaluation of SO$_4$ and BC concentrations except that OC measurements were not available for EMEP stations. The comparison of surface concentration measurements

at the IMPROVE stations (Fig. 15) shows a negative bias which may be a consequence of neglecting to explicitly compute the formation of secondary organic aerosols in this model setup, or to missing OC sources, such as marine emissions of organic species. However, since also the simulated BC aerosol has a similar negative bias it is more likely that some combustion sources that contribute to both the BC and OC concentrations are underestimated by the model. The negative bias is reduced in the GFAS simulation in which both BC and OC emissions are enhanced. The correlation R between OC model results and

observations between 0.49 and 0.57 is lower than for sulfate, where R=0.92 for IMPROVE stations alone (not shown).

For the aircraft measurements the comparison with modeled OC (Fig. 16) provides a similar picture. While still the modeled OC values are within the measurement variability indicated in the figure, for the ACE-Asia, ARCTAS (Arctic region), DODO and DABEX (both West Africa) and VOCALS (Pacific) campaigns the GFAS results clearly show higher OC concentrations compared to the NUDGE and CLIM experiments. The higher concentrations agree better with the measurements for the Arctic,

but for the African and Pacific concentrations the GFAS results overestimate the measured values. For the AMMA campaign the modeled sulfate concentrations considerably overestimate the measurements for the NUDGE and CLIM simulations, but here a good agreement is found for GFAS. For aircraft measurements in North America and Europe the model partly underestimates OC concentrations near the surface considerably, but the agreement at higher altitudes is well within the uncertainty range of the observations.

### 5.6 Mineral dust

Model results for mineral dust are compared to AOT and AE retrievals at selected AERONET stations that are dominated by dust aerosol, and dust concentrations measured at surface stations from the AEROCE and SEAREX programs. The locations of the in-situ measurements are illustrated in Fig. 17.





Modeled AOT and AE for the CLIM and NUDGE experiments are compared for AERONET stations that were labeled as 'dusty' by Huneeus et al. (2011). AOT time series for a subset of these stations are shown in Fig. 18. Overall the AOTs are higher for stations influenced by dust compared to the non-dust stations in Fig. 4, exceeding monthly mean values of 1 in multiple instances. The temporal changes from daily to interannual timescales in dust AOT are strongly controlled by the

surface wind speeds in dust source regions that lead to dust emissions if a wind speed threshold is exceeded. Therefore the monthly and interannual changes of AOT in dust-controlled regions are clearly better matched to the AERONET observations for the NUDGE compared to the CLIM simulation. This is also evident in Fig. 19 that relates monthly AOTs averaged for days when measurements were available at the respective AERONET stations. The correlation coefficient between annual AOTs for model results and observations is 0.45 and 0.61 for the CLIM and NUDGE simulations, respectively. This is expected

as the nudged meteorology should capture individual dust events better than the meteorology from the free model run. The model results have a slight negative bias indicating insufficient dust amounts. The negative bias is partly due to discrepancies in Arabian stations, where dust sources may not be sufficiently characterized. RMS (0.23) and bias (-0.14 resp. -0.16) are similar for both experiments.

The simulated AE at the AERONET stations (Fig. 20) shows a positive bias for all regions, again indicating too small

particle sizes or underestimated coarse mode dust particles in the model. Again the correlation is better when using nudged meteorology to emit and transport dust in the model compared to the free run (R values of 0.78 for the correlation of annual AEs for both the CLIM and NUDGE simulations).

Huneeus et al. (2011) performed a similar evaluation for annual averages of dust simulations by several AeroCom models. Compared to that study, the correlations of the annual average AOT and AE results and observations from the NUDGE simu-

lation are higher compared to the earlier version ECHAM5-HAM2, but slightly lower than for the AeroCom median. Huneeus et al. (2011) found correlation coefficients of R=0.23 for annual averaged AOTs for the previous version ECHAM5-HAM but as much as 0.85 for the AeroCom median. The spatial correlations of ECHAM5-HAM AE were 0.74, and 0.81 for the annual means. This is in the range of the results for the NUDGE experiment for which the correlation coefficient for annually averaged AOT is 0.61 and the AE correlation coefficient is 0.78, as stated above.

Other than for the AOT at AERONET sites with strong dust influence, the comparison of model results and measurements of monthly mean dust surface concentrations at the AEROCE and SEAREX sites (Fig. 21) shows some instances where the disagreement at some stations exceeds an order of magnitude. It should be kept in mind that for the surface concentration results - in contrast to the AERONET comparisons - the time periods of simulations and observations were different. As for AOT, the correlation coefficient R for the NUDGE simulation ist 0.64, which is again clearly better than for CLIM results with

R=0.49. The sigma values reflecting the ratios of simulated and observed variabilities at the station locations are 1.2 and 2.5 for NUDGE and CLIM, respectively. The variabilities in the model surface concentrations are higher than the observations, which is contrary to the AERONET comparisons. The annually averaged concentrations can be compared to the values for the same comparison by Huneeus et al. (2011) (Table 4). There correlations of annual averaged concentrations of 0.84 for CLIM and 0.91 for NUDGE are each higher than the previous model version ECHAM5-HAM (R=0.8) and for the NUDGE simulation



also better than the AeroCom median with R=0.82. NUDGE results also have a lower bias, RMS is however higher for CLIM and similar for NUDGE compared to the results from ECHAM5-HAM.

**Table 4.** Comparison between observed and simulated ECHAM6-HAM2 annual average dust surface concentrations at the locations of the AEROCE and SEAREX stations (see Figure 17). (*): cited from Huneeus et al. (2011).

| Data | CLIM | NUGDE | ECHAM5-HAM(*) | AeroCom median (*) |
|---|---|---|---|---|
| *Surface dust concentration* | | | | |
| Correlation coefficient | 0.84 | 0.91 | 0.80 | 0.82 |
| Absolute mean bias | 2.87 | -0.29 | -2.18 | -1.45 |
| Relative mean bias | 0.26 | 0.14 | -0.46 | -0.39 |
| RMS | 17.3 | 3.8 | 4.1 | 3.1 |
| Sigma (Mod. Stdv/Obs. Stdv) | 2.9 | 1.1 | 0.4 | 0.7 |

## 5.7 Sea salt aerosol

In ECHAM6.3-HAM2.3 the sea salt emission parameterization is changed for the model standard setup compared to previous
releases. The standard emission scheme is now based on Long et al. (2011) and includes a temperature dependence according to Sofiev et al. (2011) that was derived as parameterization from laboratory measurements. The temperature dependence my be a consequence of the temperature dependence of surface tension of the sea surface, or due to higher solubility of air entrained in the surface water at colder temperatures leading to less bubble production and thus lower sea salt aerosol emission. The temperature correction causes an increase in sea salt aerosol mass emissions fluxes in regions where sea surface temperatures
are above 20°C, and decrease at lower temperatures. At the same time number emission fluxes increase at lower and decrease at higher sea surface temperatures compared to the temperature-independent parameterization.

The results for surface concentration and size distribution are compared for sea salt emission schemes that can be selected in the HAMMOZ namelist. Compared are results from nudged simulations using the previous ECHAM-HAM default scheme by Guelle et al. (2001) (Guelle), the often used emission scheme by Gong (2003) (Gong) and a model version where the
Gong scheme is modified by the temperature dependence according to Sofiev et al. (2011) (Gong-T). The differences in the emission characteristics of the different emission schemes and their performances in a regional aerosol-transport model are shown in Barthel et al. (in review). Higher emission fluxes for particle sizes above 2 $\mu$m are expected for Guelle and Gong compared to NUDGE parameterization because in contrast to those parameterizations, spume drops contributing to those large particle sizes are not included in the Long et al. (2011) emission scheme. Thus the total emitted sea salt mass is lower
compared to parameterizations including the influence of spume drop formation. However such large particles sediment out quickly and thus have very short lifetimes so that their impact on both radiative fluxes and as CCN is expected to be small.



Simulations with the different sea salt emission schemes in ECHAM6.3-HAM2.3 are compared for the year 2010. The sea salt aerosol has only a minor influence on AOTs except over the Southern Ocean where the contribution of anthropogenic and dust aerosols is small. The model results were compared with measurements from the AERONET Maritime Network (MAN) (aeronet.gsfc.nasa.gov/new_web/maritime_aerosol_network.html, Smirnov et al. (2009)) taken on individual research cruises.

Compared are the simulated and measured AOTs for NUDGE and Guelle for the year 2006 that had good data coverage (Fig. 22a and b). While both simulations have a slight negative bias, the correlation for NUDGE is with 0.83 better that the Guelle AOT results with R=0.79. In addition, for AERONET stations in the Southern Ocean where the sea salt aerosol contribution is considerable the daily AOTs and AEs are shown in Fig. 22c and d for collocated model results. While it is evident that the AOT is better matched for CLIM, NUDGE and GFAS results compared to the Guelle results that overestimate AOTs, the AEusing

the new model sea salt emissions overestimate AE which again points toward missing coarse mode aerosols in the model due to the neglect of sea salt aerosol formed by spume droplets.

The model results were also evaluated against sea salt surface concentrations measured at the AEROCE and SEAREX stations (Fig. 17). Only stations where the sea salt concentrations remain below $100\mu/m^3$ are considered, as higher concentrations indicate local influences that cannot be captured by the model.

The scatter plots show that the temperature dependence improved the correlation between monthly measurements and model simulations (Fig. 23). Correlations are still worse than those for the dust surface concentrations as the station measurements may be influenced by local conditions not well captured by the model, but increased from R=0.18-0.19 for Guelle and Gong to to R=0.31 for NUDGE and Gong-T. The bias is negative for the temperature dependent emissions. RMS errors are similar for the different simulation results. For the time series for a subset of individual stations it can be seen that the model results mostly

stay within the error bars indicating the standard deviation of the observations (Fig. 24). Most differences are evident for the treatment of temperature in the different simulations. For stations between 45° N and 45° S the different model setups provide similar results, where no individual emission scheme performs best for all stations. For high latitude stations north of 45° N or south of 45° S the surface concentrations computed in the simulations that include a temperature dependence (NUDGE, Gong-T) clearly match the observed sea salt concentrations better than the results using the original Gong and Guelle emission

schemes without temperature correction.

Not only concentrations but also particle size dependences are influenced by the different emission formulations. As with optical depths, even the oceanic aerosol size distribution is strongly influenced by aerosols other than the sea salt aerosol, e.g. anthropogenic or natural sulfates. For the comparison with the compilation of aerosol particle size distributions at different marine sites compiled by Heintzenberg et al. (2000), only for the region 40-60°S discernible differences for the different model

results are found (Fig. 25), as only in this region the sea salt distribution has a notable impact compared to anthropogenic and biomass contributions to aerosol number size distributions in other oceanic regions. The temperature-dependent results are shifted to smaller particles sizes compared to the results from modeled sea salt emissions that do not include temperature dependence. In contrast to mass emissions, the number size concentration for accumulation mode particles is higher in the NUDGE setup using the Long et al. (2011) parameterization than for the other model results and matches best the observed number





concentrations. Considering the evaluation of both mass concentration and particle number concentration, the parameterization by Long et al. (2011) including a temperature dependence can be considered an overall reasonable choice.

## 6   Conclusions and outlook

The aerosol-chemistry-climate model ECHAM-HAMMOZ has been updated and improved since the previous release version
Zhang et al. (2012). The aerosol part ECHAM6.3-HAM2.3 is evaluated against a standard set of aerosol observations including AOT and AE from sunphotometer measurements, particle size distribution and in-situ measurements of mass concentrations of different aerosol species including aircraft measurements. A comparison against the previous results was not the main focus of this paper since both the host model ECHAM and the aerosol model have been updated at the same time. The updates of the aerosol model include changes in the model structure, bugfixes, updates in aerosol processes including updates for
aerosol water uptake, and cloud activation, and updated aerosol emissions. Anthropogenic emissions of $SO_2$, OC and BC from ACCMIP and biomass burning emissions from ACCMIP or GFAS datasets can be chosen. Emissions of mineral dust now include updated Saharan dust sources, and allow for coupling with the JSBACH land surface scheme. A regional tuning parameter was introduced to account for changes in the surface parameterization of the ECHAM model. This is expected to become obsolete in a future improved dust emission scheme. A new sea salt aerosol emission scheme was implemented
that includes a temperature dependence of sea salt emission fluxes. The aerosol model can be used in combination with the chemistry module in the ECHAM-HAMMOZ setup (Schultz et al., 2017) or with a simplified sulfur chemistry, which is evaluated in this publication. The alternative aerosol setup with the sectional aerosol scheme ECHAM6.3-HAM2.3-SALSA was evaluated by Kokkola et al. (2018).

The model performs well in the comparison of the different aspects of the aerosol distribution. Using state-of-the-art anthro-
pogenic aerosol emissions is the basis for investigations examining the role of anthropogenic aerosol changes in the climate system. Attention must also be given to carefully characterizing natural aerosol distributions. Mineral dust and sea salt aerosol distributions must be well characterized in order to be able to make meaningful statements on the importance of anthropogenic aerosol effects.

As natural aerosol distributions are strongly impacted by dust and sea salt particle emissions, particular attention was given
to updating and testing these aerosol species. In the new version of ECHAM6-HAM2 they compare more favorably to observations than in the previous version. However, due to the description of the aerosol size distribution by modes, large particle sizes may be underestimated, which is evident in the overestimate of the AE in regions dominated by dust and sea salt aerosol. While neglecting part of the coarse mode particle load may have only a minor influence on the particle number and thus CCN concentrations , mass fluxes may be underestimated. A positive bias in the comparison of AE may also point towards an under-
estimate in coarse mode aerosols emitted by biomass burning. Overall the model reproduces AOTs and sulfate concentrations in US and European sites well, but to some extent underestimates BC and OC concentrations.

As expected, the model versions using nudged wind fields (NUDGE) to simulate atmospheric aerosol transport (and emissions in the case of mineral dust and sea salt) perform better in terms of reproducing the temporal variability in aerosol





distributions at different timescales compared to the free (CLIM) runs. However, differences in bias and variabilities of the CLIM and NUDGE simulations are small.

Even where the evaluation of the aerosol distributions simulated with the updated ECHAM6.3-HAM2.3 model show only small improvements compared to earlier model versions and discrepancies remain as e.g. in the underestimation of BC and OC concentrations, the use of more realistic aerosol processes and updated emissions are prerequisite for reliable model studies of the effects and interactions of aerosols in the climate system.

Further evaluation with monitoring and field data will be performed in ongoing projects. Further developments in the model will include updates in the secondary aerosol scheme and adding nitrate aerosol to the microphysics scheme.

*Code availability.* The ECHAM6-HAMMOZ code is maintained and made available to the scientific community under https://redmine.hammoz.ethz.ch/. The availability is regulated under the HAMMOZ Software Licence Agreement that be downloaded from https://redmine.hammoz.ethz.ch/attachments/download/291/License_ECHAM-HAMMOZ_June2012.pdf.

*Data availability.* AERONET data can be obtained with the Aerosol Robotic Network download tool https://aeronet.gsfc.nasa.gov/cgi-bin/webtool_opera_v2_new. MODIS products are available for download from Level 1 and Atmosphere Archive and Distribution System (LAADS) https://ladsweb.modaps.eosdis.nasa.gov/search/. EMEP data is available for download at http://ebas.nilu.no/. IMPROVE data is available for download from the Federal Land Manager Environmental Database hhttp://views.cira.colostate.edu/fed/DataWizard/Default.aspx. AEROCE and SEAREX data can be downloaded from http://aerocom.met.no/download/DUST_BENCHMARK_HUNEEUS2011. The BC aircraft measurement data are available at http://aerocom.met.no/download/BC_BENCHMARK_KOCH2009/. EUSAAR size distributions can be downloaded from https://doi.pangaea.de/10.1594/PANGAEA.861856. The aircraft data for sulfate and OC was received from several measurement teams who hold the ownership for the data.

*Competing interests.* The authors declare that they have no competing interests

*Acknowledgements.* The ECHAM-HAMMOZ model is developed by a consortium composed of ETH Zurich, Max Planck Institut for Meteorology, Forschungszentrum Julich, University of Oxford, the Finnish Meteorological Institute and the Leibniz Institute for Tropospheric Research, and managed by the Center for Climate Systems Modeling (C2SM) at ETH Zurich. The research leading to these results has received partly funding from the Center for Climate System Modelling (C2SM) at ETH Zurich and European Union's Seventh Framework Programme (FP7/2007-2013) project BACCHUS under grant agreement no. 603445. This work was supported by a grant from the Swiss National Supercomputing Centre (CSCS) under project ID s652. We are grateful for computing time from the Swiss Computing Centre (CSCS) and from ETH Zurich, and for the Deutsches Klimarechenzentrum (DKRZ). Computing resources at DKRZ were granted under project number bb1004. P.S. acknowledges funding from the European Union's Seventh Framework Programme (FP7/2007-2013) projects



BACCHUS under grant agreement 603445 and the European Research Council project ACCLAIM under grant agreement FP7280025 as well as the European Research Council project RECAP under the European Union's Horizon 2020 research and innovation programme with grant agreement 724602. HK acknowledges support by the Academy of Finland project no. 308292 and 307331, Nordforsk project no. 57001.The data used are listed in the references and are available under https://redmine.hammoz.ethz.ch. We thank the AERONET principal

5      investigators and their staff for establishing and maintaining the sites used in this manuscript. IMPROVE is a collaborative association of state, tribal, and federal agencies, and international partners. We thank Maria Kanakidou (ECPL, University of Crete) for the help in compiling the EMEP and IMPROVE datasets. For the aircraft data we thank Colette Heald (MIT, Dept. Civil and Environmental Engineering), Hugh Coe (University of Manchester), Lynn Russell (Scripps Institution of Oceanography), Rodney Weber (Georgia Institute of Technology), Jose Jimenez (University of Colorado at Boulder), Roya Bahreini (University of Colorado - CIRES, NOAA ESRL Chemical Sciences Division),

10     Ann Middlebrook (NOAA ESRL Chemical Sciences Division), James S.McDonnell Foundation Award for 21st Century Science, NOAA grant NA17RJ1231, National Science Foundation grants ATM-0002035, ATM-0002698, and ATM04-01611, and the NERC Global Aerosol Synthesis and Science Project (GASSP) NE/J023515/1. IMPROVE is a collaborative association of state, tribal, and federal agencies, and international partners. US Environmental Protection Agency is the primary funding source, with contracting and research support from the National Park Service. The Air Quality Group at the University of California, Davis is the central analytical laboratory, with 20 ion analyses

15     provided by Research Triangle Institute, and carbon analysis provided by Desert Research Institute.





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





**Figures**

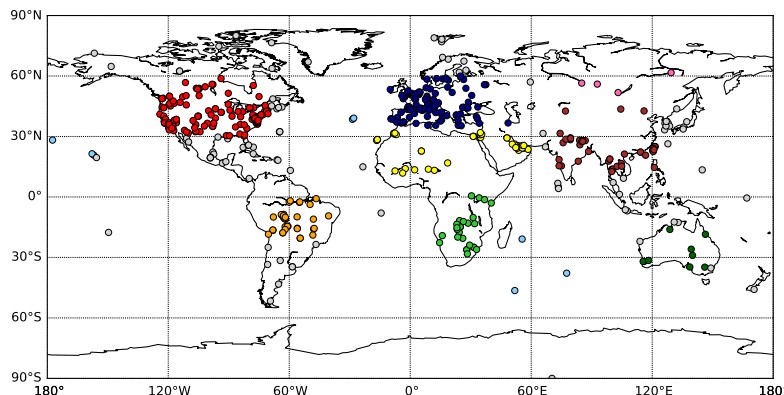

**Figure 1.** Locations of AERONET stations used for model evaluation. All stations are color coded according to the region to which they belong (Red: North America; Dark blue: Europe; Brown: East Asia; Pink: Siberia; Yellow: North Africa; Green: South Africa; Orange: South America; Dark green: Australia; Light Blue: Oceanic regions.)





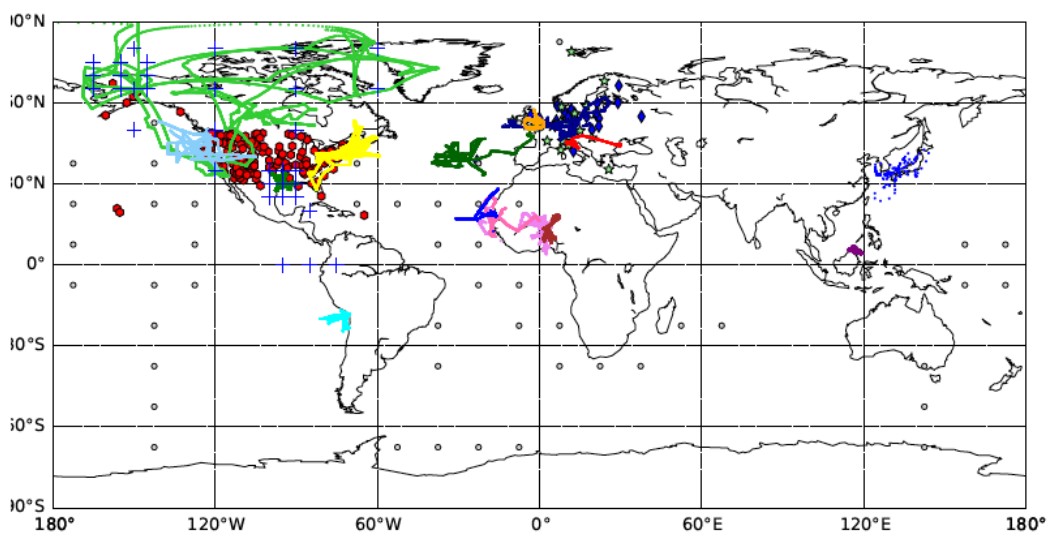

**Figure 2.** Networks of surface stations and research aircraft flight tracks used for model evaluation. Blue diamonds: EMEP stations with sulfate concentrations; Red hexagons: IMPROVE stations with concentrations of sulfate, BC, and OC). Continuous color lines: aircraft flights for the evaluation of sulfate and OC vertical profiles (Heald et al., 2011); Blue crosses: regions for the evaluation of BC vertical profiles (Koch et al., 2009). Green stars: European sites with size distributions (Asmi et al., 2011); Grey circles: oceanic regions with size distributions (Heintzenberg et al., 2000).





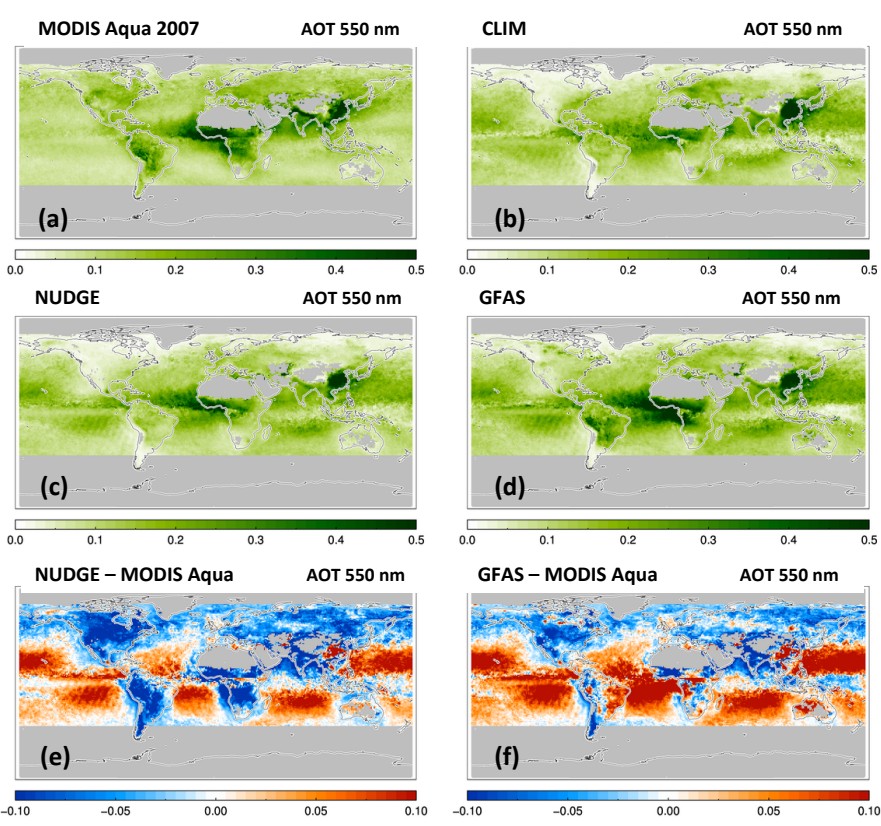

**Figure 3.** Comparison annual average aerosol optical thickness (AOT) retrieved from MODIS Aqua satellite measurements (a) and for the experiments CLIM, NUDGE and GFAS (b-d) for the year 2007. Additionally differences between the simulated annual average of collocated AOT and the MODIS retrievals are given for the CLIM (e) and NUDGE (f) model results.





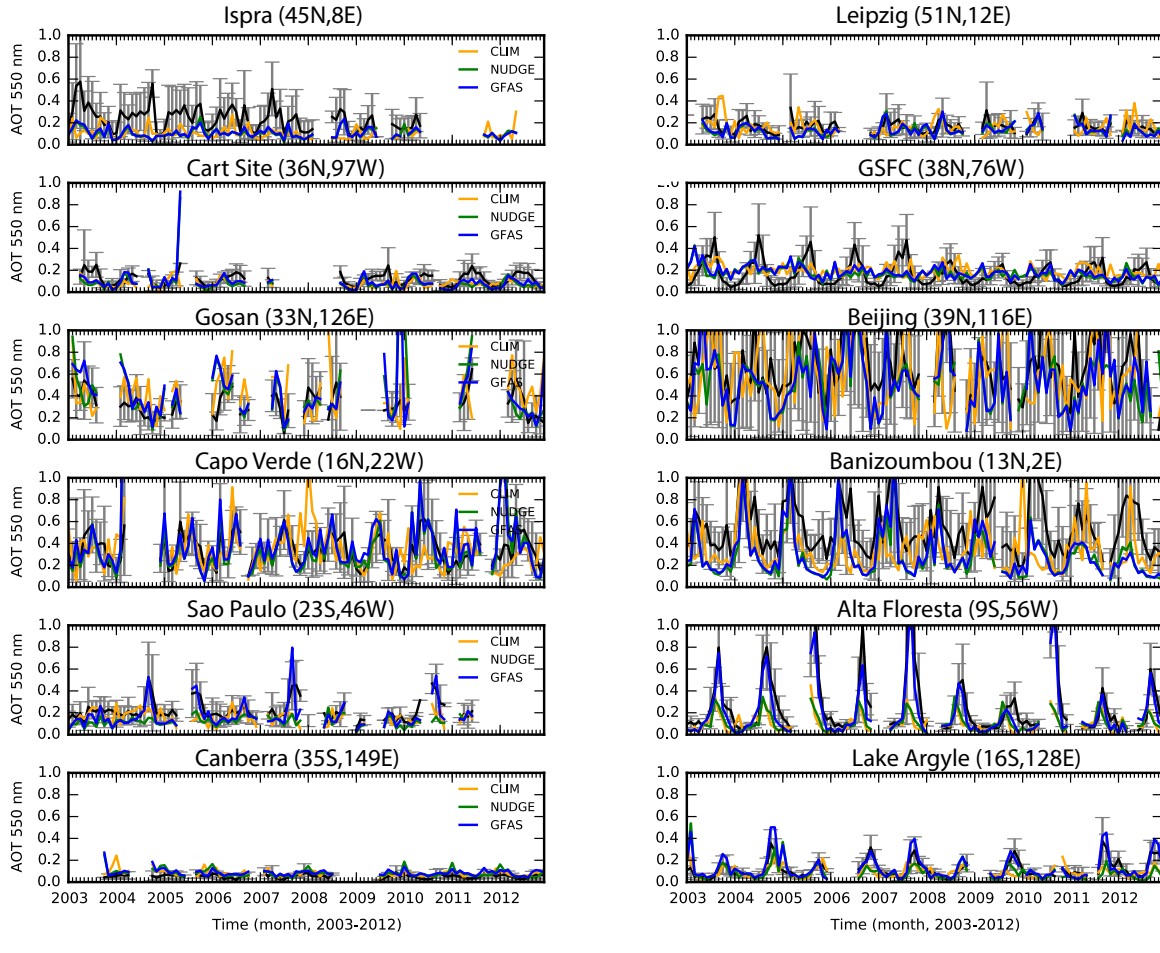

**Figure 4.** Time series (monthly means) of observed and simulated AOT (black line) from Jan 2003 to Dec 2012 at selected AERONET stations. Simulated monthly mean were constructed from the daily mean outputs sampled on the same days of the observations and collocated to the observation position. Error bars show the variabilities of the measurements. Compared are model results for the CLIM (orange) and NUDGE (green) and GFAS (blue) simulations.





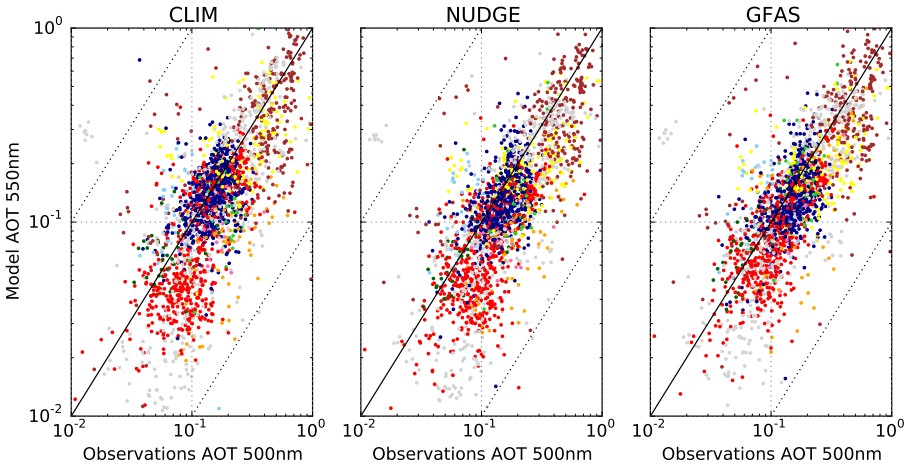

**Figure 5.** Scatterplots of observed versus simulated mean AOT over the period Jan 2003 to December 2012 at AERONET stations shown in Figure 1. The simulated yearly means are constructed by sampling the model from daily mean outputs for the same days of observations and collocated to the locations of the observations. Stations are color coded depending on the regions to which they belong to as shown on Figure 1. Red: North America; Dark blue: Europe; Brown: East Asia; Pink: Siberia; Yellow: North Africa; Green: South Africa; Orange: South America; Dark green: Australia; Light Blue: Oceanic regions.

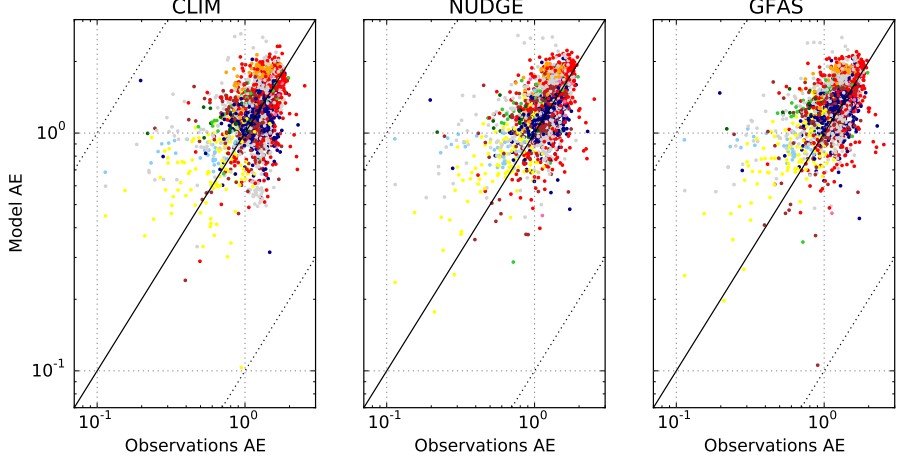

**Figure 6.** Scatterplot of observed versus simulated mean Ångstrom exponents over the period Jan 2003 to December 2012 at AERONET stations shown in Fig.1 for the CLIM, NUDGE and GFAS simulations. The color coding of the results are identical to and the simulated means were constructed as in Fig. 5.

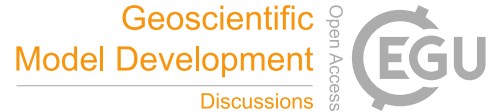

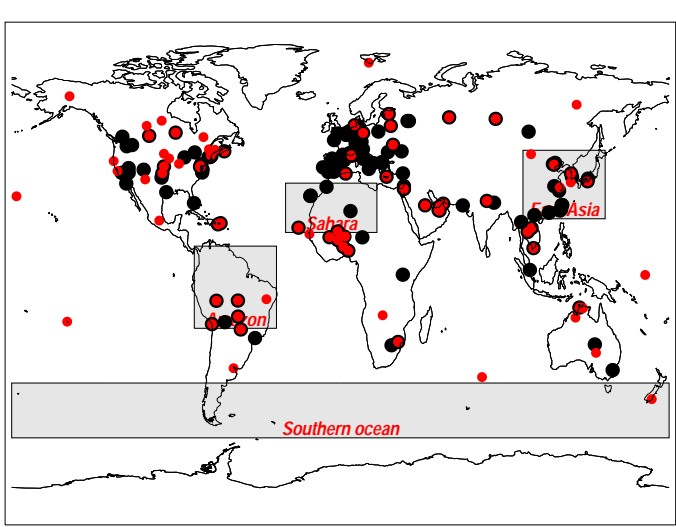

**Figure 7.** AERONET stations and regions used in the monthly summaries for AOT, AE (direct sun measurements, red symbols) and SSA (version 2 inversion product, (Holben et al., 2006), black symbols) for year 2007 in Fig. 8. The stations for the AOT and AE summaries are selected as being regionally representative as in Kinne et al. (2013).





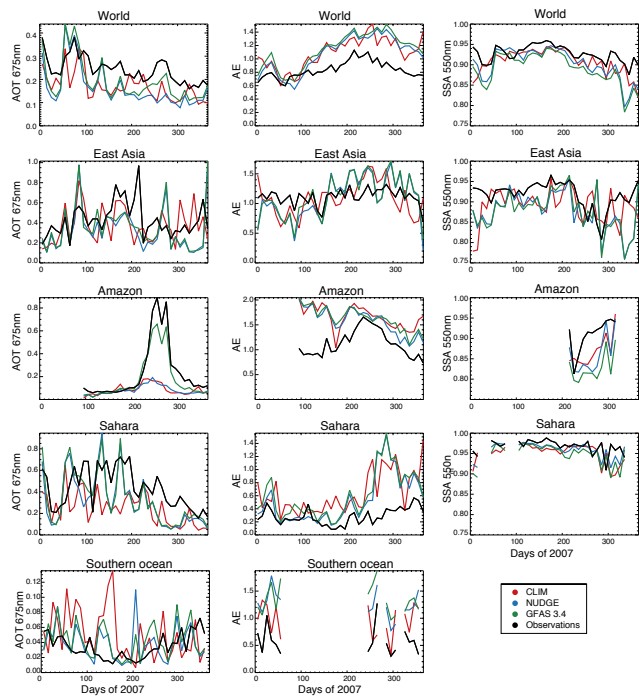

**Figure 8.** Annual cycle of AOT (left panels), AE (middle panels) and SSA (right panels) from AERONET retrievals for global averages and summarized for several regions (top-to-bottom panels: World, East Asia, Amazon, Sahara, Southern oceans) as shown in Fig. 7 for the year 2007.





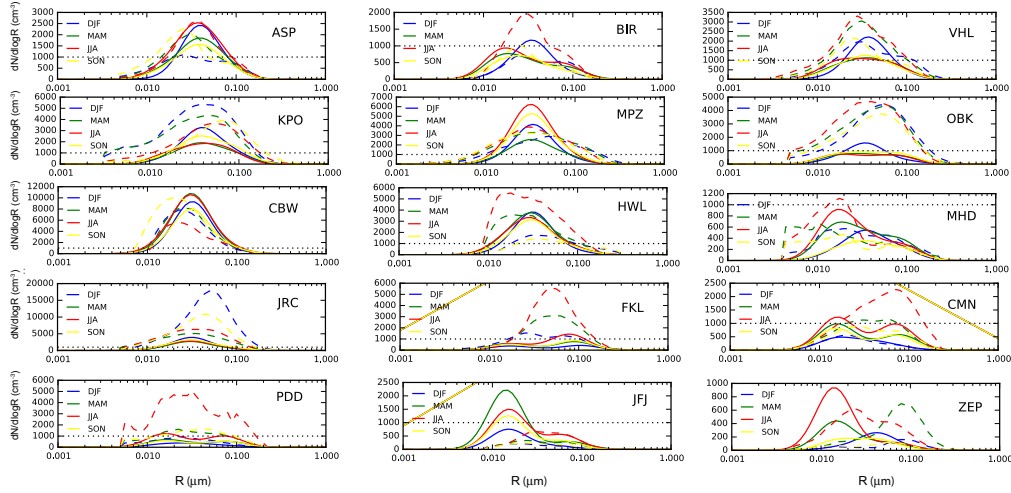

**Figure 9.** Observed (dashed lines) and simulated (solid lines) (NUDGE simulation) near-surface median aerosol size distributions for European field monitoring sites (EUSAAR) for 2009 (Asmi et al., 2011). Data are for winter (blue), spring (green), summer (red), fall (yellow). The simulated size distributions are the median of the number of daily mean size distributions per season. The top panel contains the data for three nordic and Baltic stations (ASP: Aspvreten; BIR: Birkenes; VHL: Vavihill), the second row contains Central European sites (KPO: K.Puszta; MPZ: Melpitz,; OBK: Kosetice), the third row Western European stations (CBW: Cabauw; HWL : Harwell; MHD: Mace Head); the fourth row: stations in Mediterranean countries (JRC: Ispra; FKL: Finokalia; CMN: Monte Cimone) and the fifth row: high altitude (PDD: Puy de Dome; JFJ: Jungfraujoch) and Arctic (ZEP: Zeppelin) stations.



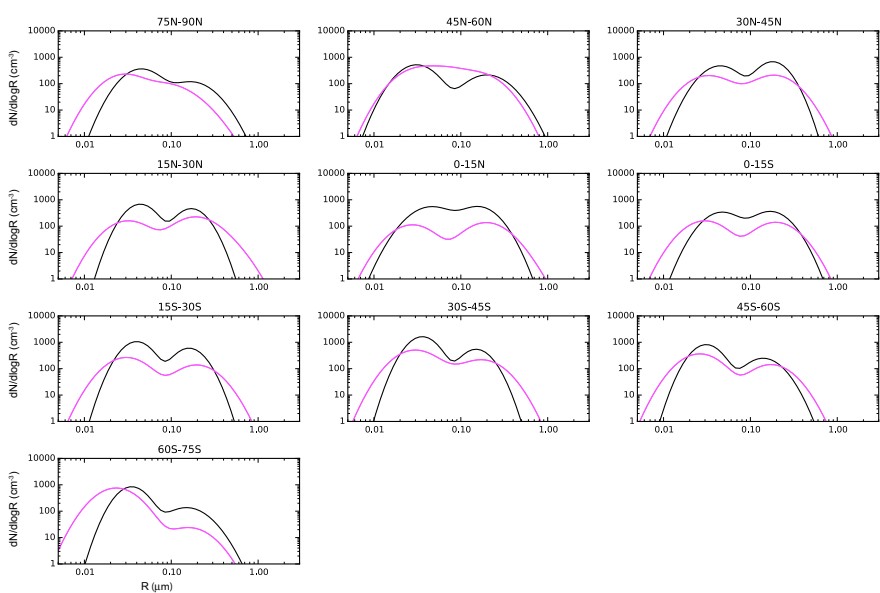

**Figure 10.** Size distribution of simulated (pink lines) and measured (black lines) aerosol number in the marine boundary layer for the NUDGE simulation. The observed size distribution corresponds to a 30-yr climatology for the Aitken and accumulation modes (soluble and insoluble) (Heintzenberg et al., 2000). The simulated size distributions correspond to an 10-year annual average for over locations of the measurements and zonally averaged between the given latitude bounds.



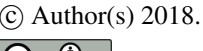

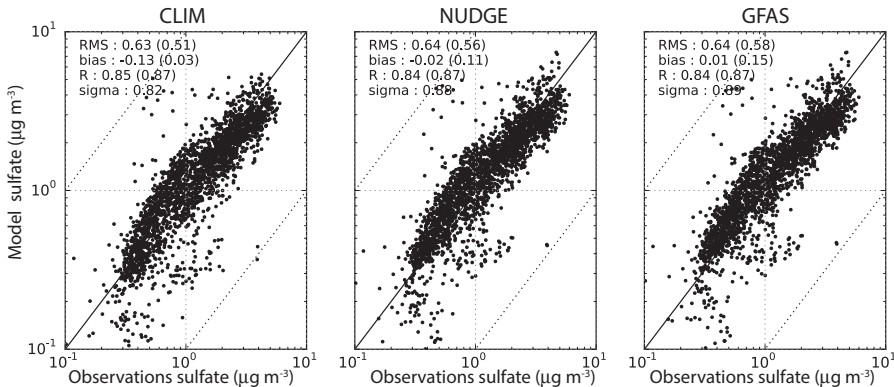

**Figure 11.** Scatter plot of sulfate surface concentrations from the EMEP and IMPROVE networks and from the CLIM (first column), NUDGE (second column), and GFAS (third column) simulations. Model data were selected for days when observations were available at each station location and yearly averaged. Observed and collocated simulated averages for all available stations (see Fig. 2 for the location of the stations) were compared for the years 2003 to 2012 for EMEP stations and 2000-2004 for Improve stations. For each comparison the root mean square error RMS (normalized RMS in parenthesis), the correlation coefficient R (R on log scale in parenthesis), the absolute bias (normalised bias), and the ratio between simulated and observed standard deviation (sigma) are given.





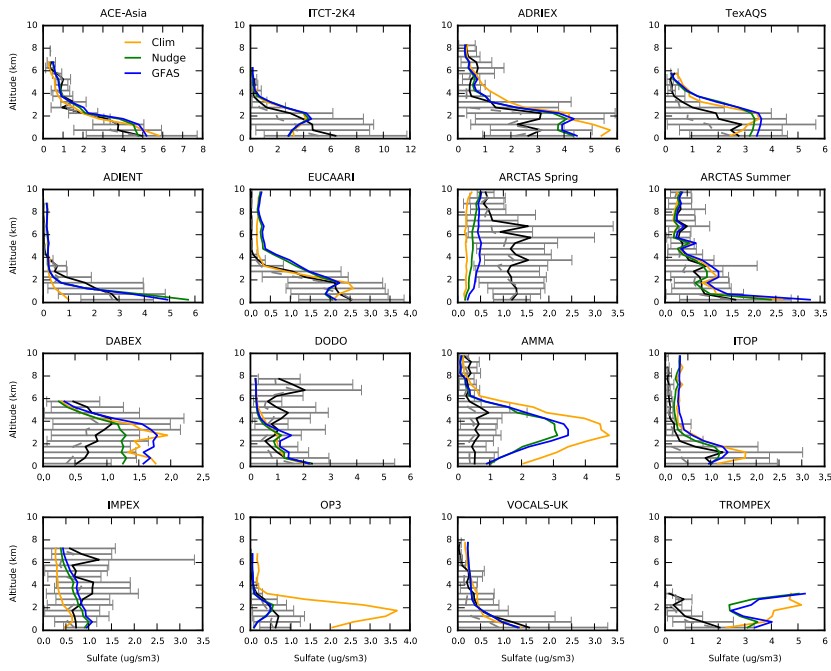

**Figure 12.** Vertical profiles of observed concentrations of sulfate (black) and simulated concentrations from the three simulations, CLIM (orange), NUDGE (green), and GFAS (blue). The observations are provided for 16 aircraft campaigns that investigated different regions of the world from 2001 to 2009 (Heald et al., 2011) (see also Fig. 2). The model is sampled along the flight tracks using a temporal average of the outputs over the duration of the campaign, and the error bars show the variabilities in the measurements. Simulated vertical profiles are shown as monthly and regional averages.





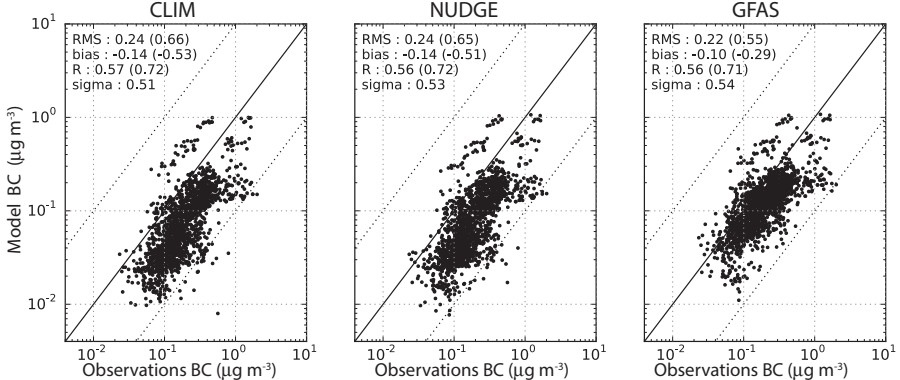

**Figure 13.** As Fig. 11 for black carbon (BC) aerosol.

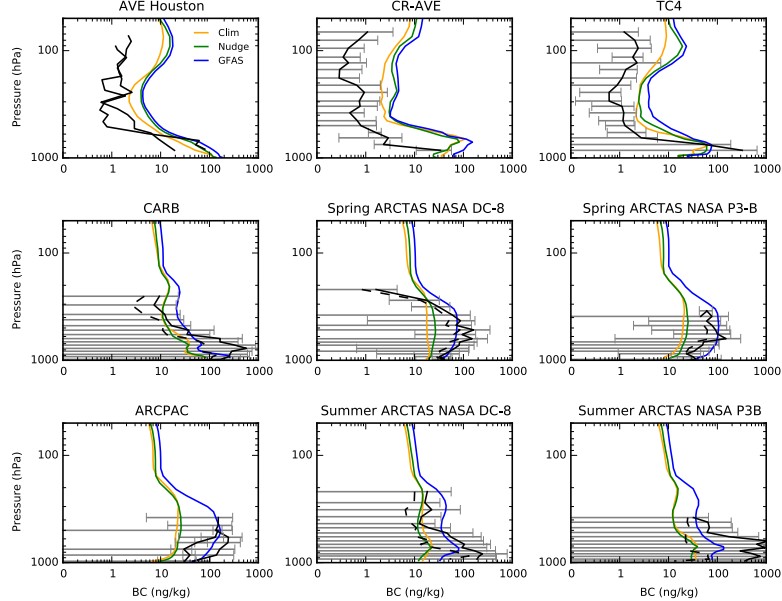

**Figure 14.** Vertical profiles of observed concentrations of BC (black) and collocated simulated concentrations from the three simulations, CLIM (orange), NUDGE (green), and GFAS (blue). The observations are provided for 9 locations and seasons (see Fig. 2) (Koch et al., 2009). Observations are averaged for the respective campaigns (standard deviations are provided where available) and mean (solid black) and median (dashed black) profiles are shown for some campaigns. Model outputs (monthly averages) are sampled over specific points in each region.



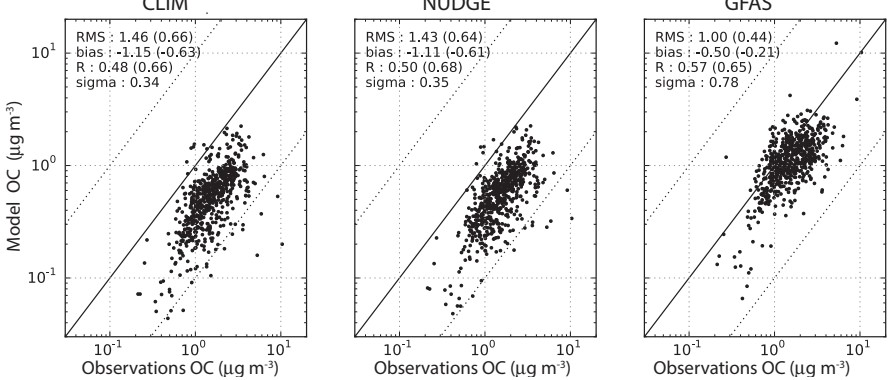

**Figure 15.** Scatter plot of observed surface concentrations of OC from the IMPROVE network and collocated simulated daily concentrations from the CLIM (first column), NUDGE (second column), and GFAS (third column) simulations. The yearly mean is calculated from January 2000 to December 2004 for all available stations (see Fig. 2 for the location of the stations). The statistical parameters are calculated as in Fig. 5.

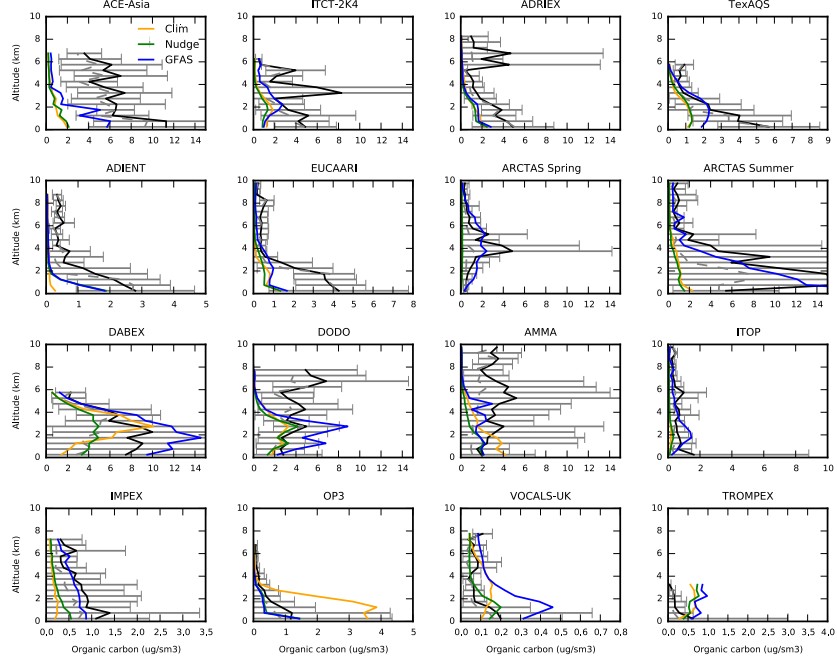

**Figure 16.** As Fig. 12 for OC aerosol.





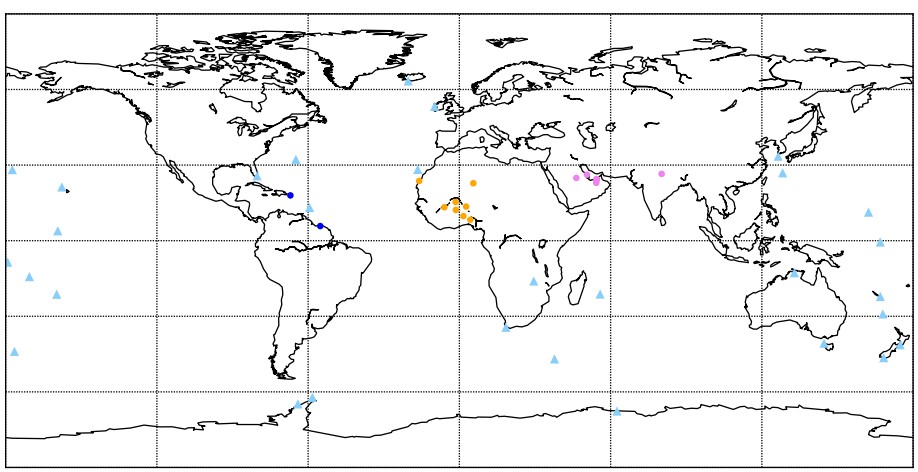

**Figure 17.** Locations of stations used for the evaluation of dust (DU) and sea salt (SS) aerosol. Triangles: AEROCE and SEAREX stations with dust and sea salt surface measurements. Circles: AERONET stations labelled as "dusty" by Huneeus et al. (2011). Yellow: North Africa; Pink: Middle-East and Asia; Dark Blue: Central America, Light Blue: marine stations.





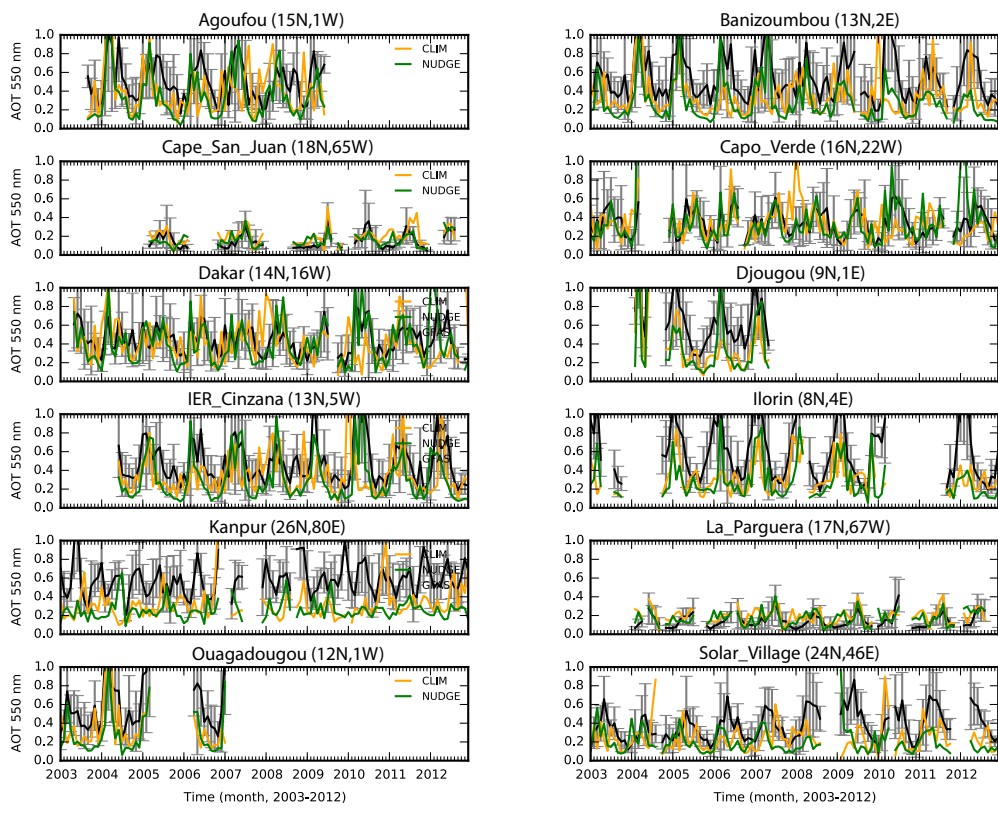

**Figure 18.** Time series of AOT at selected stations labelled as "dusty" by Huneeus et al. (2011) for the years 2003 to 2012 for the CLIM and NUDGE simulations.




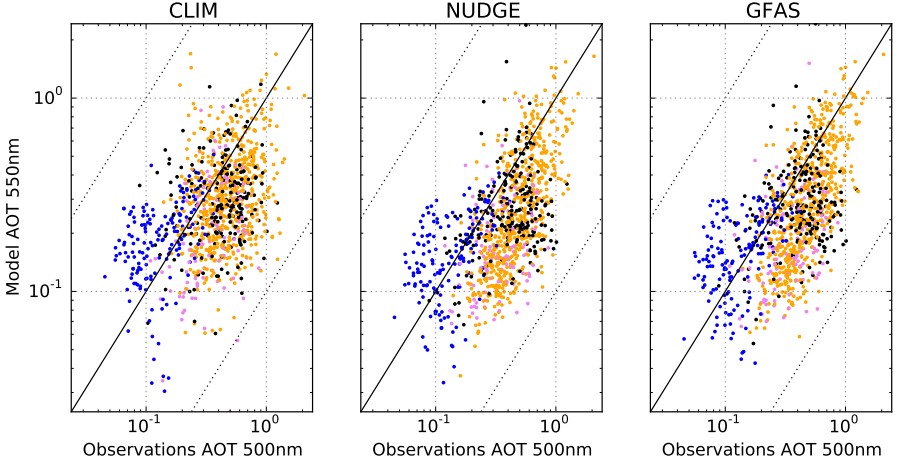

**Figure 19.** Scatterplot of observed versus simulated monthly mean dust AOT based on daily results at AERONET stations shown in Figure 17. The simulated monthly means are constructed by sampling the collocated model from daily outputs for the same days as the observations. Stations are color coded depending on the regions to which they belong to as shown in Figure 17. Yellow: North Africa; Pink: Middle-East and Asia; Dark Blue: Central America, Light Blue: marine stations.

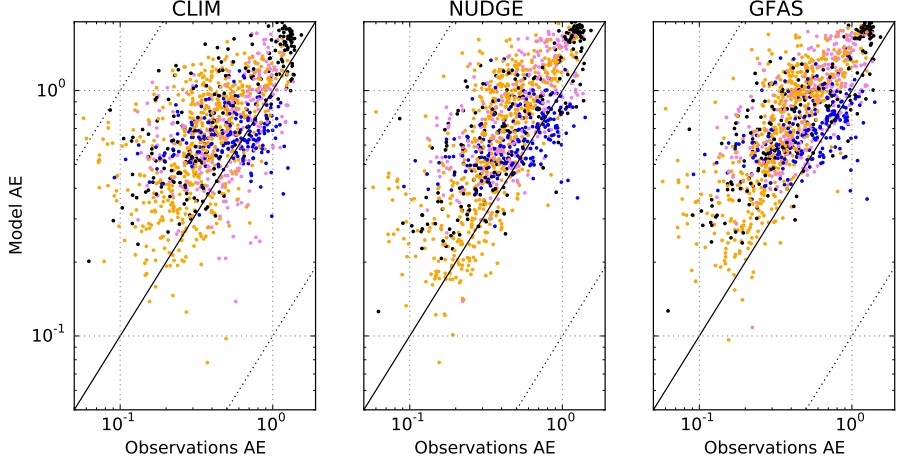

**Figure 20.** As Fig. 19 for the Ångstrom exponent.





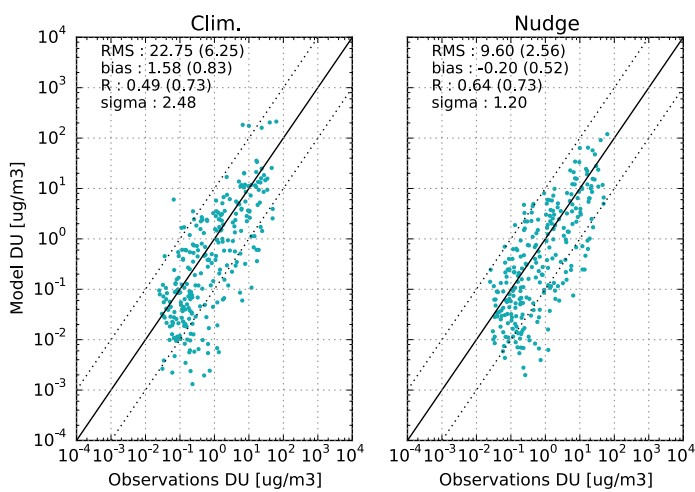

**Figure 21.** Scatterplot of observed versus simulated monthly mean dust surface concentrations at AEROCE and SEAREX stations shown in Fig. 17. Simulated monthly mean were constructed from the daily mean outputs sampled on the same days of the observations and collocated to the observation position for the time period 2003-2012.





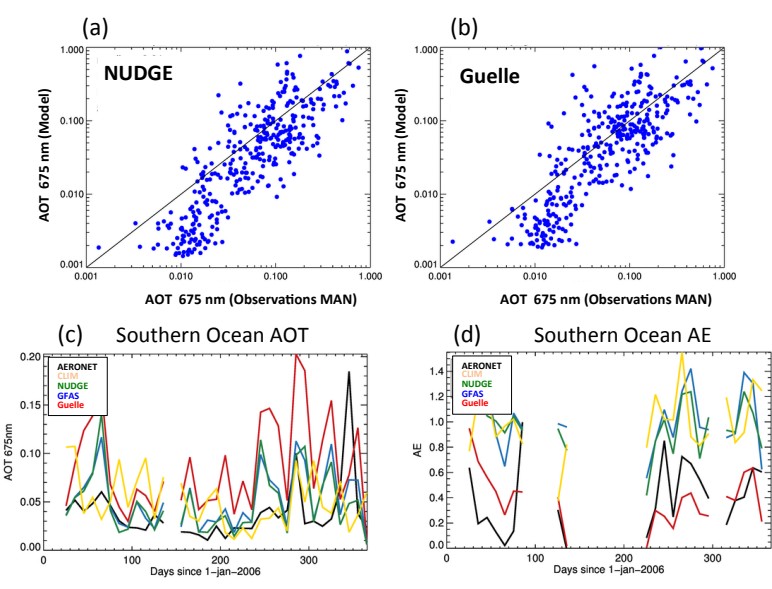

**Figure 22.** Comparison of model results with sunphotometer data for (a) AOT at stations of the AERONET Maritime Aerosol Network for NUDGE simulations (2007) (b) as (a) for simulations with the Guelle sea salt emission parameterization; (c) Time series for year 2007 comparing AOT from AERONET stations for NUDGE, CLIM, GFAS and Guelle simulations in the Southern Ocean; (d) as (c) for AE)





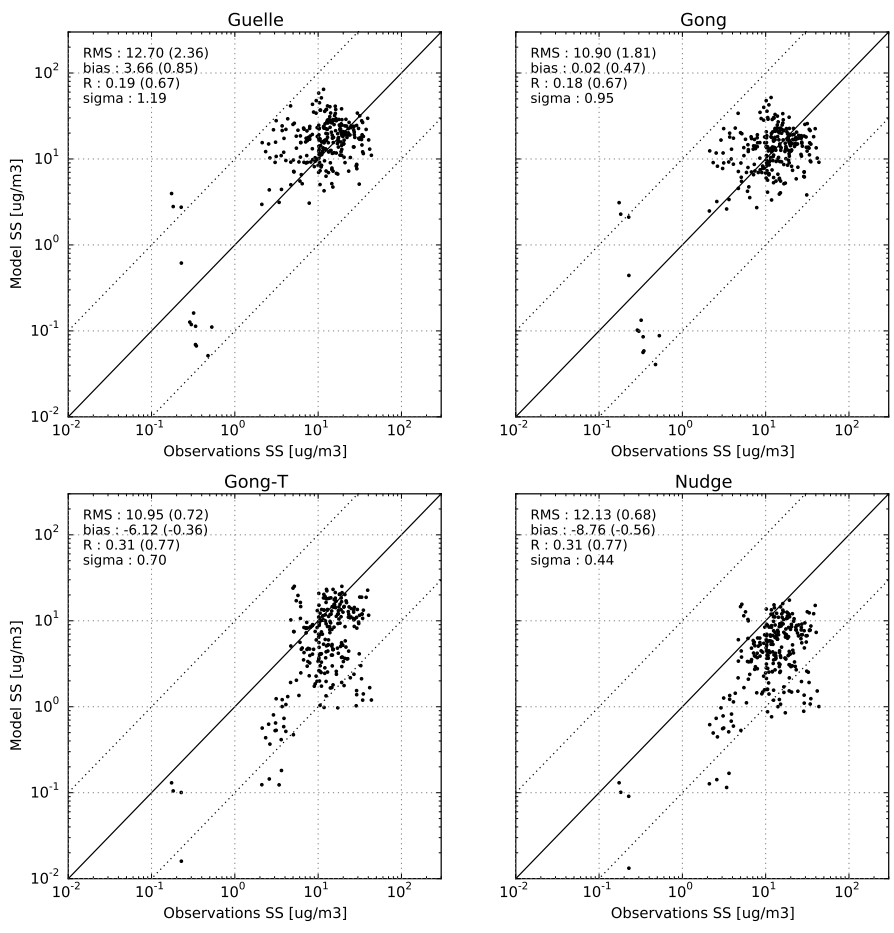

**Figure 23.** Scatterplot of observed versus simulated monthly mean sea salt surface concentrations at AEROCE and SEAREX stations shown in Fig. 17 for year 2010. Compared are simulations using different sea salt emission parameterizations (Guelle, Gong, Gong-T, NUDGE).





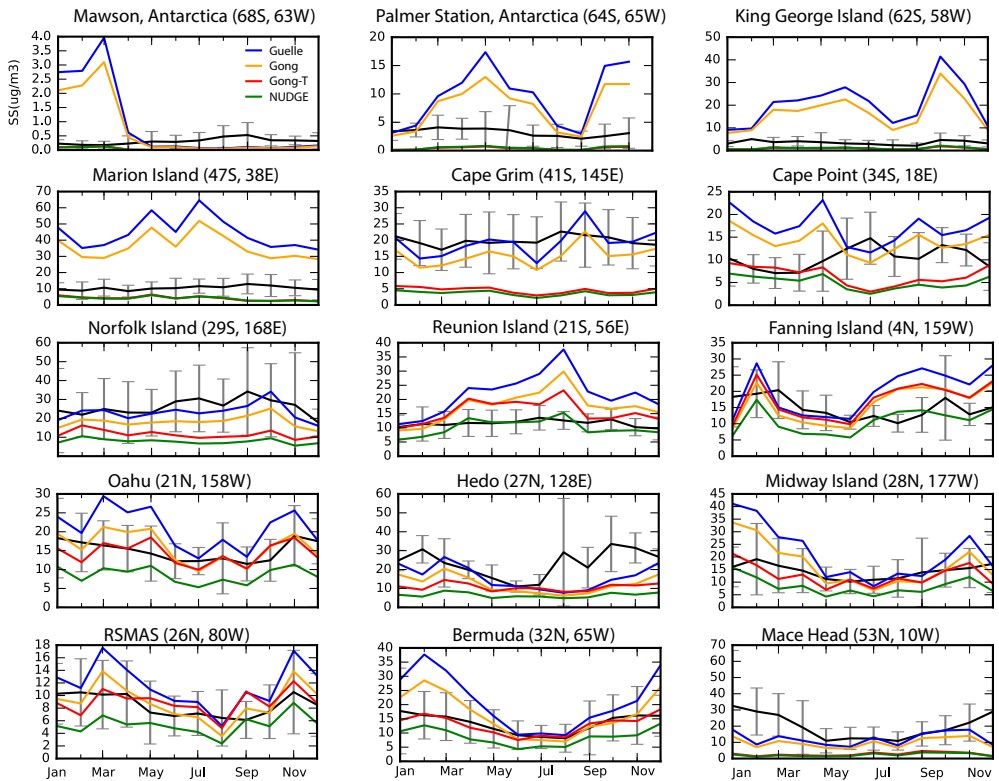

**Figure 24.** Time series of observed versus simulated monthly mean sea salt aerosol surface concentrations for the year 2010 spatially collocated at the AEROCE stations shown in Fig. 17. Only stations where the sea salt concentrations remain below $100\mu$ m$^{-3}$ are considered to exclude stations with clearly local impact.



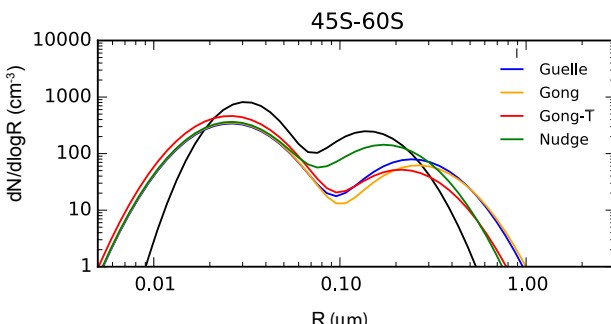

**Figure 25.** Size distribution of simulated and measured aerosol number concentrations in the marine boundary layer for the region 40-60° S (as in Fig. 10) for the year 2009.