# Peer review of "The global aerosol-climate model ECHAM6.3-HAM2.3 – Part 1: Aerosol evaluation"

_Geoscientific Model Development, 2018_

## Referee Comment (RC1) · Anonymous Referee #1 · 19 Nov 2018

General comments:

The present study shows an update version of ECHAM-HAM model with considerable effort to improve the schemes and verify results. The manuscript is well written, and it will be acceptable for publication. However, I have two concerns about the present study. The first is a sea salt. The simulated sea salt results are greatly changed. The global burden and emission flux in the updated version are significantly reduced with previous version and references, as shown in Table 3. The physical process in the updated version may be improved, but the simulated sea salt results are not well improved. It is much better to add evidence of sea salt improvement to the present manuscript. Another point is that the evaluation of the simulated number concentrations is a bit rough. Quantitative evaluation using statistic metrics is necessary to verify the

results.

Specific comments:

P.6, L9-11: Which species are emitted from a model level corresponding to a specific altitude?

P.7, L26-31: Is the correction factor a tunable parameter?

P.11, Table 2: Are all tuning parameters set to the same among these three experiments?

P.12, Section 4.3: Why are the authors not using measurements in Asia? EANET (http://www.eanet.asia/index.html) can be useful for the model evaluation of sulfate.

P.13, L17-19: In section 5.5.2 and 5.5.3, the authors pointed out the possibility of the underestimation of OC and BC emitted from combustion sectors. This can be also mentioned here.

P.13, Section 5.2: Do the authors compare AOT with AERONET under the clear sky conditions? Or whole sky? Please clarify it.

P.14, L10-19 (Figure 5): Could you add the statistical metrics to Fig 5 or new table?

P.16, L5: The authors insist "Overall the agreement is good in most cases", but there is no quantitative discussion. Could the authors add some statistic metrics to this analysis? Also, how about the error bar of the measurements?

P.19, L1-3: As I mentioned in general comments, the uncertainty of the emission fluxes over the United States is expected to be relatively low, but the authors mention the possibility of the lack of local emission. Do the authors have any evidence?

P.21, Table 4: The values are different form that shown in Figure 21. Why?

P.21, Section 5.7: As I mentioned in general comments, the improvement of the module for seasalt is very interesting, but the evaluation may be inadequate and should be

more validated. For example, how about the AOT over the Southern Ocean? In addition, how about the evaluation of fields for cloud and precipitation, which can strongly affect the scavenged process?

P.24, L4-6: The issue of the model grid size can be also important to resolve such local emission. Do the authors have a plan to use finer horizontal resolution?

Figure 3: Why is the region (>40S) in the Southern Ocean shaded? I think MODIS can capture these regions, even though the uncertainty may be large.

Figure 11: IMPROVE also measures 2003-2012, but why do the authors select the specific years of 2000-2004?

Figure 19: How about the statistic metrics like the correlation coefficient?

Technical corrections:

P.6, L33: I think the resolution is 0.1 degree in the latest GFAS (version 1.2), but do the authors use the older version?

P.7, L1-4: In Veira et al. (2015), the scaling factor is recommended in version 1.1. Now, the latest version is 1.2. So, do the authors use GFAS version 1.1? Please clarify it.

P.18, Table 3: It is much better to add the range of AeroCom estimates.

Figure 1: What is the grey circle?

Figure 2: What is the purple in Kalimantan Island? What is the blue around Japan?

Figure 5: What is the grey?

Figure 9: Some panels (FKL, CMN and JFJ) include yellow lines. Please remove the lines.

Figure 10: How about the error bar of the measurements?

Figure 19: dust AOT → AOT in the dusty region?

Figure 22: In the panels (c) and (d), which station is it?

**[GMDD](https://doi.org/10.5194/gmd-2018-235)**

Interactive
comment

---

## Referee Comment (RC2) · Anonymous Referee #2 · 22 Nov 2018

The manuscript "The aerosol-climate model ECHAM6.3-HAM2.3: Aerosol evaluation" by Tegen et al. provides a documentation of the new version of the aerosol module in the ECHAM model, and a comprehensive evaluation of aerosol simulation. They use in-situ surface and aircraft measurements and satellite data to evaluate different aspects of the model, finding that the model generally perform very well. This is an excellent paper for its scientific and technical significance as well as its clear and concise presentation. I strongly recommend this paper to be published as soon as possible.

I have a few comments for the authors to consider.

In Table 3 and 4 you provided the comparison between this model with AEROCOM and the previous version of the model. I find the comparison very useful! Please consider to do similar comparison for the rest of the paper, when possible, either by provid-

ing the values/plots from the previous version, or by providing a couple of sentences and citation to the papers documenting the evaluation of the previous model. It might also be a good idea to provide your experience on what change(s) in the model (from ECHAM5-HAM2 to ECHAM6.3-HAM2.3) cause the specific improvement. This kind of information can help other modeling centers to make similar improvements. Page 7, Line 12-31: Could you please provide a few sentences to describe what might be the cause(s) for the need of using different scale factors in different regions. Is it because the surface wind bias in different regions are different? Or, is it because the satellite-based dust map is insufficient to constrain the emission? Does model resolution play a role here? Lastly, how do you decide the scale factor? Do you use dust AOD from satellite or in-situ measurements? Page 8, Line 20-26: Since SOA is often considered poorly treated in GCMs, could you please briefly describe the uncertainty of the SOA treatment and its potential impact on the conclusion. Providing some literature review will be great. Page 10, Line 13-25: When pressure and winds are nudged, do you need to retune the model either to restore the TOA energy balance and/or to restore the reasonable surface flux (of heat, moisture, momentum, and sea salt and dust)? If the CLIM run's surface winds and NUDGE run's surface winds are very different, surface fluxes will be significantly different. Also, what is the relaxation time scale for nudging? Page 13, Line 13-16, The 2 nudged simulations produce very large AOD bias over subtropical ocean (Fig. 3). This might indicate that the sea salt emission needs to be tuned down. Section 5.3: I think near the source region the size distribution comparison is not very interesting since the model's size distribution is closely related to the assumed size distribution at emission. The comparison between more interesting when it is done for remote regions when the model physics has enough time to change the size distribution. I wonder if it is possible to do the analysis that way. Could you please comment? Figure 5, 6, 19, 20, 22a,b: Please also add the R, RMSE, and mean bias in the figure.

---

## Referee Comment (RC3) · Miller (Referee) · 30 Nov 2018

This is a well-written article that represents an impressive amount of work to collect a wide range of observations for evaluation of an aerosol model. I have some suggestions for improving clarity, but the article should be suitable for publication subject to minor revision. If the authors have any questions, they can contact me at ron.l.miller@nasa.gov.

1. The article is a detailed and extensive evaluation of the current model. The article could have even more lasting value if it anticipated and aided development of future model versions.

My main suggestion is to add more quantitative metrics to certain figures. These met-

rics can be useful when future generations of the aerosol model are developed and evaluated. For example, Figures 19 and 20 will be of limited use in future development, because the current model assessment is mainly visual and qualitative (i.e. are the points clustered around the 1:1 line?). These figures would benefit from the summary statistics that are listed in Figures 21 or 23 (e.g.), so that similar figures of future model versions can be compared using these statistics. This suggestion also applies to Figures 5, 6 along with 22a and b. Figures 4, 8, 18 and 24 could also be improved with summary metrics. More generally, the authors should think about how they might assess future model versions and augment the figures so that any eventual improvement can be quantified.

The article would also benefit from a discussion in the conclusions about what the authors consider to be key model errors that should guide future development.

Technical Comments (page, line number)

2, 18: add citation to Huneeus et al. ACP 2011?

2, 30: somewhere in this paragraph, it would be helpful to state explicitly that the fields calculated by the chemistry module MOZ are prescribed in this study, so that only the aerosol calculation is fully interactive. (I suggest this distinction because Section 2 describes the capabilities of both the HAM and MOZ modules: a joint description that I like for placing the HAM module in context.)

3, 1: add 'or bin' after 'sectional'?

Section 2.2 ('ECHAM'): it would be helpful if the horizontal and vertical resolution of the model version used in this study was listed here, rather than postponed to Section 3.

Table 1 caption: 'Mode boundaries for the ' missing word?

5, 20: 'parameterization for *ocean* temperature dependence'?

6, 8: 'individual sectors'? Define or give examples of 'sectors' here (instead of in the

following paragraph)?

6, 21: 'T63' This is one reason why the ECHAM resolution would be more helpfully defined in Section 2.2 than Section 3.

6, 27: 'Interannual variability of biomass burning is not considered.' But is the decadal-averaged burning updated each year via interpolation so that it doesn't change abruptly each decade?

7, 13: 'Dust particle emissions are driven by...' How is the ratio of emitted silt particle sizes to clay sizes determined? This is relevant to the coarse-mode burden. which later is found to be unrealistically small.

7, 19: 'Tegen et al. (2002), who identified potential source areas...' It's worth noting that the sources are also identified using a calculation of paleo-lake extent (that I consider innovative).

7, 30: 'These regional correction factors...' What observations and criteria were used to arrive at this correction?

8, 5: 'As a marine source...' What physical variables control emission of DMS? (wind speed? ocean temperature?)

Section 2.3.2 ('Aerosol Microphysics') The omission of nitrate aerosols should be noted, since this could contribute to an underestimate of PM or AOT.

9, 11: 'in-cloud scavenging scheme' Observed scavenging depends upon whether the aerosol is hydrophilic, and this will vary with aerosol species (with sulfates being very hydrophilic and some common dust minerals as hydrophobic). Does droplet activation account for varying aerosol composition within each size mode?

9, 20: '1.52 + 0.0011' It should be noted that this represents relatively little shortwave absorption by dust, in better agreement w/ AERONET retrievals (Sinyuk et al GRL 2003) than the original Patterson et al. (1977) laboratory measurements of far-traveled

[Figure]

Saharan dust samples.

9, 27: Does dust nucleate ice particles by enabling heterogeneous freezing? Does nucleation vary with the aerosol mixture in each size mode, given that some aerosols like dust are much more efficient?

10, 14: 'are relaxed' What is the time scale for relaxation? Does it vary with height?

10, 21: 'do not vary on daily or interannual time scales' But is emission updated each year to reflect slow decadal trends, or is emission held constant over the simulation period?

10, 22: After 'satellite retrievals' insert '(labeled "GFAS")'?

11, 9: 'respective AERONET stations' How many years of measurements are available for the stations? Do some stations have short records that may not be representative of the simulation decade?

11, 14: 'data-assimilation grade product based on Dark Target retrievals' What are the advantges of this product compared to the standard off-the-shelf version of MODIS AOT?

11, 20: 'and with compiled number size distributions...by Heintzenberg et al. (2000)' Maybe move this phrase down as few sentences to where this measurement set is discussed in detail?

12, 7: IMPROVE. it should be noted that some of these sites are at elevation in regions where the topography is not well-resolved by the model. Also, why did you not evaluate dust using these measurements? e.g. see:

VanCuren, R., and T. Cahill, Asian aerosols in North America: Frequency and concentration of fine dust, J. Geophys. Res., 107(D24), 4804, doi:10.1029/2002JD002204, 2002.

13, 5: Why restrict the comparison of MODIS AOT to 2007? Is this year representative

of the entire simulation?

13, 17: 'This may point to missing aerosol sources' Maybe say 'aerosol species' instead of aerosol 'sources'? Also, Bauer et al. (2015) argue that nitrate aerosols and ammonium sulfate represent a significant fraction (a little more than half) of the anthropogenic contribution to PM2.5 over central North America. The omission of these aerosol species from HAM could contribute to the underestimate of AOT.

Bauer, S. E., K. Tsigaridis, and R. Miller (2016), Significant atmospheric aerosol pollution caused by world food cultivation, Geophys. Res. Lett., 43, 5394–5400, doi:10.1002/2016GL068354.

Figure 4 caption: 'Error bars show the variabilities of the measurements.' How is the variability defined? Standard deviation of daily values?

14, 16: 'slightly'? 0.01 seems like a large improvement compared to -0.3.

Figure 8: Could labels be added to the top of each column explaining the quantity plotted (AOT, AE and SSA)?

15, 16: 'underestimating the particle size' Is it known why this discrepancy over the Sahara is largest in the NH Fall?

15, 22: 'the too low particle size ... could result in too high SSA' Why would coarse particles reduce the SSA? Is there a simple explanation or reference for this?

15, 34: 'too strong mixing' vertical or horizontal or both?

Figure 10 caption: replace 'for over' with 'over'?

16, 6: 'may not be representative' This is especially true for stations where topography is poorly resolved.

16, 26: Table 3 includes median values from AeroCom. It should be stated that this median is derived from models and is not necessarily in agreement with observations.

Also, this comparison between HAM2.3 and AeroCom would be more useful is some measure of model diversity was added to the AeroCom column of Table 3.

16, 30: 'However...emission fluxes also depend on the size range considered...' This is a very relevant caveat. To me, it suggests that the comparison of HAM and AeroCom emission be replaced by a comparison of aerosol burden.

16, 34: 'in agreement with the AeroCom average burden of 19.2 Tg' Again, the AeroCom burden is not necessarily indicative of the observed value. As an alternative, Kok et al. 2017 calculate a global burden of around 20 Tg that is better constrained by observations:

Kok, J.F., D.A. Ridley, Q. Zhou, R.L. Miller, C. Zhao, C.L. Heald, D.S. Ward, S. Albani, and K. Haustein, 2017: Smaller desert dust cooling effect estimated from analysis of dust size and abundance. Nature Geosci., 10, no. 4, 274-278, doi:10.1038/ngeo2912.

17, 14: 'R on log scale' Does this mean that R is calculated by correlating logarithms of concentration and not concentration itself? This is fine, but it should be explained that such a correlation emphasizes the ability of a model to correctly simulate variations in concentration over large distances from the source, rather than subtle differences over more limited regions.

Figure 12 caption: 'Simulated vertical profiles are shown as monthly and regional averages.' What is the duration of the flight data sets that are compared to the monthly averaged model concentration in this figure?

Table 3 caption: does 'sedimentation flux' refer to gravitational settling?

19, 8: 'while the CLIM and NUDGE results underestimate BC' The values from these simulations are indeed smaller than the GFAS values, but all seem to be with in the undertainty of the measurements.

20, 15: 'the correlation is better' ... 'R values of 0.78 for the correlation of annual AEs for both the CLIM and NUDGE simulation' If the correlations are the same, how can

one be better?

21, 6: 'my be' (may be?) Why 'may be'? Doesn't Soffiev attribute the temperature dependence to specific physical processes like surface tension and solubility (or is the temperature dependence entirely empirical)?

21, 18: 'spume drops' Could you define spume drops and their relation to the emitted size distribution for the benefit of readers like myself :) who are not specialists in sea-salt aerosols?

page 22: the years of comparison for sea salt are confusing. Line 1 cites the year 2010. But line 5 claims that 2006 is used in Figure 22, whose caption says 2007. In fact, Figure 23 does seem to be based on 2010 according to the caption, but this is not noted in the text (line 15 onward).

22, 22: 'For high latitude stations ...' This generalization doesn't seem to be true for the only NH high latitude station. Also, why are the NUDGE and Gong-T models with better physics behaving so poorly at Cape Grimm (41 s), where temperature effects might be expected to be important? The improved agreement at Marion Island, just 6 degrees poleward, suggests individual stations might include large regional effects that are unique to each island.

23, 21: 'Mineral dust and sea salt aerosol distributions must be well characterized in order to be able to make meaningful statements on the importance of anthropogenic aerosol effects.' I agree with this, but could you justify this point more fully? For climate change, we don't need to characterize natural sources if their radiative forcing is time-independent. To be sure, dust sources expand and contract with time (e.g. due to variations in hydroclimate). Nonetheless, it is hard to evaluate simulated anthropogenic species if the natural species are poorly simulated, because common observed variables like AOT or PM2.5 include the effect of all species together. (Thus, most Aero-Com models get reasonable values of AOT despite widely varying fractions of natural and anthropogenic species.)

---

## Author Comment (AC1) · 23 Jan 2019

Reviewer 1:

General comments:
The present study shows an update version of ECHAM-HAM model with considerable effort to improve the schemes and verify results. The manuscript is well written, and it will be acceptable for publication. However, I have two concerns about the present study. The first is a sea salt. The simulated sea salt results are greatly changed. The global burden and emission flux in the updated version are significantly reduced with previous version and references, as shown in Table 3. The physical process in the updated version may be improved, but the simulated sea salt results are not well improved. It is much better to add evidence of sea salt improvement to the present manuscript. Another point is that the evaluation of the simulated number concentrations is a bit rough. Quantitative evaluation using statistic metrics is necessary to verify the results.

*Thank you for the positive assessment. The discussion of the modeled changes in sea salt emissions and burdens and the assessment of size distributions are explained in the detailed comments below.*

Specific comments:

P.6, L9-11: Which species are emitted from a model level corresponding to a specific altitude?

*While anthropogenic and wind blown species are emitted at the surface, species emitted from biomass burning (BC, OC, SO$_2$) are emitted with 75% evenly distributed in the planetary boundary layer, 17% in the first level and 8% in the second level above the PBL. This is explained further down in the text where emissions of individual species are described.*

P.7, L26-31: Is the correction factor a tunable parameter?

*The correction factor can be modified in the model for each region via the model namelist and should be checked for different model resolutions. We had omitted that in the case of nudged simulations the regional correction factors are different.*
*We modified the text: 'For each relevant region that contains dust sources the correction factors are chosen such that the emissions agree with the values by Huneeus et al. (2011). These regional correction factors can be modified via the model namelist. For this model version they are set to 1.45 for North- and South America and Asia, and 1.05 for all other regions for the simulations that were not nudged. For the nudged simulations the correction factors are 1.25 for North- and South America and Asia, and 0.95 for all other regions. '*

P.11, Table 2: Are all tuning parameters set to the same among these three experiments?

*Regional dust tuning parameters are different between NUDGE and CLIM, see above.*

P.12, Section 4.3: Why are the authors not using measurements in Asia? EANET (http://www.eanet.asia/index.html) can be useful for the model evaluation of sulfate.

*Thank you for pointing out those data. Within EANET wet deposition measurements are compiled, while elsewhere we compare model results with surface concentration measurements. Here we did not use deposition measurements for model evaluation. Interpretation of these comparisons may be problematic as they are strongly depending on local precipitation variability (which is not necessarily captured by the model due to the coarse resolution). However a focused investigation of the wet deposition performance of the model would be an interesting future study.*

P.13, L17-19: In section 5.5.2 and 5.5.3, the authors pointed out the possibility of the

underestimation of OC and BC emitted from combustion sectors. This can be also mentioned here.

*Done*

P.13, Section 5.2: Do the authors compare AOT with AERONET under the clear sky conditions? Or whole sky? Please clarify it.

*The comparisons of model AOT were done with the cloud-screened version 2 of the AERONET data, representing clear-sky conditions. In the model the clear-sky AOT was calculated based on clear sky relative humidity. The comparisons were for the same time period for the model and the measurements.*

P.14, L10-19 (Figure 5): Could you add the statistical metrics to Fig 5 or new table?

*Statistical information was added for figures 5,6,19,20 and 22.*

P.16, L5: The authors insist "Overall the agreement is good in most cases", but there is no quantitative discussion. Could the authors add some statistic metrics to this analysis? Also, how about the error bar of the measurements?

*The discussion of the comparison of the compilation of marine aerosol size distribution data is expanded in section 5.3. Regarding the error bars of the measurements, the aerosol size distribution dataset compiled by Heintzenberg et al. (2000) that is the basis for this comparison has been composed of 30-year measurements from 18 different datasets in marine regions. From those data they derived in addition to aerosol number concentrations the geometric mean diameters and standard deviations of the Aitken and accumulation modes assuming lognormal mode size distributions. Individual errors can thus not be given because uncertainty ranges are not provided. For this reason statistic metrics are not provided.*

P.19, L1-3: As I mentioned in general comments, the uncertainty of the emission fluxes over the United States is expected to be relatively low, but the authors mention the possibility of the lack of local emission. Do the authors have any evidence?

*This was stated as a possibility for an explanation for the relatively low concentrations simulated in the United States. Local dust emission fluxes in the US may be too low in the model since anthropogenic dust sources are not included in this model version. Also, biomass burning emissions may be low. A key uncertainty is likely the emission size distribution representative for grid-box sizes. However there is in fact no concrete evidence to support this*

P.21, Table 4: The values are different form that shown in Figure 21. Why?

*In Figure 21 the correlations are based on monthly average dust values while the values in Table 4 are correlations of annual average values between modeled and observed dust concentration values, which can be compared to the values published by Huneeus et al. (2011). The averaging times are mentioned in the captions of the figure and table, respectively.*

P.21, Section 5.7: As I mentioned in general comments, the improvement of the module for seasalt is very interesting, but the evaluation may be inadequate and should be more validated. For example, how about the AOT over the Southern Ocean? In addition, how about the evaluation of fields for cloud and precipitation, which can strongly affect the scavenged process?

*We agree that the sea salt emission parameterization is a major change in the model. We should point out that it is easily possible to use different sea salt parameterizations in the model via a namelist switch. The major reduction of emission mass fluxes is due to the omission of spume droplets, while the temperature dependence used from Sofiev et al. (2011) is based on laboratory experiments. Further AOT evaluation has been tried, but is problematic as due to low sea salt AOT pure sea salt signals could not be distinguished even in the southern ocean or at coastal and island AERONET sites. Higher AOTs from biomass burning, some dust events and secondary aerosols including sulfate from DMS emission over the southern oceans can mask the AOT signal from sea salt aerosol. Comparison for the AERONET maritime network are provided in Figure 22. An approach to evaluate sea salt aerosol in the future will be utilizing retrievals of vertically resolved aerosol extinction from lidar measurements in marine locations, as with such measurements it would be possible to attribute aerosol extinction in marine boundary layer in remote marine areas to sea salt aerosol. As far as such measurements would be available, this should be part of a future dedicated model evaluation study. For now we refer to the publication by Barthel et al (2019), Atmospheric Environment, who evaluated the performance of different sea salt emission parameterizations (including the temperature-depended parameterization used here) in a regional model with different surface concentration datasets including size-resolved impactor measurements at the Cape Verde islands. The regional model allows a better description of the meteorological factors such as wind speeds and precipitation as it would be possible with the global model study. While those results showed that there was no optimal fit to the different measurements, the parameterization used here including emission parameterization by Long et al (2011) with the temperature dependence performed well.*
*Concerning the last point, an evaluation of clouds and precipitation in ECHAM6.3-HAM2.3 is performed in a companion study by Neubauer et al (submitted to GMDD, 2019).*

P.24, L4-6: The issue of the model grid size can be also important to resolve such local emission. Do the authors have a plan to use finer horizontal resolution?

*Some improvement for the comparison of model results and observations may be expected for higher horizontal model resolution, e.g. considering the subgridscale variability in hygroscopic particle growth. Another major effect of a better model resolution can be expected for resolving the wind systems leading to more accurate emission of primary natural aerosol. In particular this would affect mineral dust, for which the emission both depends on a threshold and it is highly non-linearly related to the surface wind speeds. We plan an update of the dust emission scheme to eliminate the need for regional emission tuning. At this point, also an evaluation of the resolution dependence of the dust emission and the other aerosol distributions in the model will be performed.*

Figure 3: Why is the region (>40S) in the Southern Ocean shaded? I think MODIS can capture these regions, even though the uncertainty may be large.

*Due to the overall very low aerosol optical thicknesses in the southern ocean high latitudes and the resulting uncertainties in the retrievals the comparisons were omitted for this region.*

Figure 11: IMPROVE also measures 2003-2012, but why do the authors select the specific years of 2000-2004?

*Apologies, the figure was mislabeled, this is now corrected. For $SO_4$ and BC the comparisons were performed for data 2003 to 2012 for both EMEP and IMPROVE stations (it was correct in the main text).*

Figure 19: How about the statistic metrics like the correlation coefficient?

*Statistical information was added for figures 5,6,19,20 and 22.*

Technical corrections:

P.6, L33: I think the resolution is 0.1 degree in the latest GFAS (version 1.2), but do the authors use the older version?

*Indeed here the GFAS version 1.0 is used which is based on 0.5$^o$ resolution. We added the version information in the text.*

P.7, L1-4: In Veira et al. (2015), the scaling factor is recommended in version 1.1. Now, the latest version is 1.2. So, do the authors use GFAS version 1.1? Please clarify it.

*Thank you for this hint. In these model simulations the GFAS version 1.0 is used. This information is now added to the text.*

P.18, Table 3: It is much better to add the range of AeroCom estimates.

*The values for standard deviations of the AeroCom results as provided in Textor et al (2006) were added to the table.*

Figure 1: What is the grey circle?

*The grey circles indicate mixed or coastal values that cannot be uniquely attributed to certain aerosol sources, e.g. oceanic regions that are strongly impacted by mineral dust or biomass burning downwind from continents. The omission of this explanation was corrected in the figure caption. We noted that for the Pacific some of the grey circles should actually be labeled as oceanic stations. We changed that attribution here and in figures 5 and 6.*

Figure 2: What is the purple in Kalimantan Island? What is the blue around Japan?

*Both are from aircraft campaigns compared by Heald et al (2011): the purple line near Kalimatan refers to OP3, the blue line at Japan to ACE Asia. For brevity we only referred to the Heald et al paper (figure 1 therein) for the locations of the aircraft campaigns (this was made more specific in the revised figure caption). Here we labeled the map to attribute the lines to the campaigns used in the model evaluation to the specific aircraft campaigns.*

Figure 5: What is the grey?

*See above answer regarding Figure 1, the plot colors correspond to the colors provided on the map with the grey circles correspond to coastal regions or regions where the aerosol is influenced by different sources. Figures 5 and 6 are updated to change some of the 'grey' stations to oceanic, see above.*

Figure 9: Some panels (FKL, CMN and JFJ) include yellow lines. Please remove the lines.

*Done*

Figure 10: How about the error bar of the measurements?

*The dataset compiled aerosol size distribution data by Heintzenberg et al. (2000) that is the basis for this comparison has been composed of 30-year measurements from 18 different datasets in marine regions. From those data they derived in addition to aerosol number concentrations the geometric mean diameters and standard deviations of the Aitken and accumulation modes assuming lognormal mode size distributions. Individual errors can thus not be given, uncertainty ranges are not provided.*

Figure 19: dust AOT ! AOT in the dusty region?

*The reviewer is correct; it should be AOT in dusty regions. The figure caption is corrected accordingly.*

Figure 22: In the panels (c) and (d), which station is it?

*This information is provided in Figure 7.*

Reviewer 2

The manuscript "The aerosol-climate model ECHAM6.3-HAM2.3: Aerosol evaluation" by Tegen et al. provides a documentation of the new version of the aerosol module in the ECHAM model, and a comprehensive evaluation of aerosol simulation. They use in-situ surface and aircraft measurements and satellite data to evaluate different aspects of the model, finding that the model generally perform very well. This is an excellent paper for its scientific and technical significance as well as its clear and concise presentation. I strongly recommend this paper to be published as soon as possible.
I have a few comments for the authors to consider.

*Thank you for the positive assessment and the suggestions.*

In Table 3 and 4 you provided the comparison between this model with AEROCOM and the previous version of the model. I find the comparison very useful! Please consider to do similar comparison for the rest of the paper, when possible, either by providing the values/plots from the previous version, or by providing a couple of sentences and citation to the papers documenting the evaluation of the previous model. It might also be a good idea to provide your experience on what change(s) in the model (from ECHAM5-HAM2 to ECHAM6.3-HAM2.3) cause the specific improvement. This kind of information can help other modeling centers to make similar improvements.

*This is a good idea, unfortunately it is problematic in this work. Zhang et al (2012) provide direct comparisons between HAM1 and HAM2.0 results for the ECHAM5-HAM setup. As mentioned in the introductory part both HAM was updated from version 2.0 and ECHAM was updated to version 6.3 simultaneously. Due to the close linkages between ECHAM and HAM e.g. in the aerosol-cloud interactions the changes in HAM cannot be separately evaluated vs. the changes in the climate model. Thus it would not be possible to attribute specific changes to the new HAM version (which is the topic of this publication). For example, changes in dust emission are strongly impacted by modifications in the ECHAM surface parameterization, such that the modifications in the HAM emissions cannot be evaluated separately.*

Page 7, Line 12-31: Could you please provide a few sentences to describe what might be the cause(s)

for the need of using different scale factors in different regions. Is it because the surface wind bias in different regions are different? Or, is it because the satellite based dust map is insufficient to constrain the emission? Does model resolution play a role here? Lastly, how do you decide the scale factor? Do you use dust AOD from satellite or in-situ measurements?

*The underlying problem that requires regional correction factors for the dust emission is the change in the surface parameterization as a consequence of coupling the atmospheric model ECHAM with the land surface scheme JSBACH. In the version ECHAM6.3 the roughness length definition was updated. This setting was adapted to ECHAM6.3-HAM2.3. With the changed roughness parameterization different regions became active dust sources compared to the previous version, as the threshold velocities for dust and surface wind shear patterns were modified. This problem is shortly addressed in Section 2.3.1. The scale factor is chosen such that the emissions from specific regions agree with the emissions from Huneeus et al (2011). The need for regional tuning is not ideal and will be addressed in future updates of the dust emission scheme.*

Page 8, Line 20-26: Since SOA is often considered poorly treated in GCMs, could you please briefly describe the uncertainty of the SOA treatment and its potential impact on the conclusion. Providing some literature review will be great.

*In this work we do not use the explicit secondary aerosol scheme but use instead the standard parameterization based on the assumption that a fixed percentage of natural terpene emissions at the surface that form SOA is directly emitted as OC in the model. This is clarified in the text. Since here the secondary aerosol formation is not explicitly treated we find this not to be a good place for a more in-depth discussion of this topic. It is however added as priority topic in the outlook section. Currently the ECHAM-HAM-SALSA model setup (Kokkola et al 2018, GMD) contains a SOA based scheme and will be more discussed and tested in-depth in forthcoming publications. See also Stadler et al, 2018, (Stadtler, S., Kühn, T., Schröder, S., Taraborrelli, D., Schultz, M. G., and Kokkola, H.: Isoprene-derived secondary organic aerosol in the global aerosol–chemistry–climate model ECHAM6.3.0–HAM2.3–MOZ1.0, Geosci. Model Dev., 11, 3235-3260, https://doi.org/10.5194/gmd-11-3235-2018, 2018.) for further discussion of SOA formation in the ECHAM-HAMMOZ model system. It may be noted that in the AeoCom publication by Tsgaridis et al (The AeroCom evaluation and intercomparison of organic aerosol in global models, ACP, 2014) both models using simple and complex organic aerosol schemes underpredicted organic aerosol concentrations.*

Page 10, Line 13-25: When pressure and winds are nudged, do you need to retune the model either to restore the TOA energy balance and/or to restore the reasonable surface flux (of heat, moisture, momentum, and sea salt and dust)? If the CLIM run's surface winds and NUDGE run's surface winds are very different, surface fluxes will be significantly different. Also, what is the relaxation time scale for nudging?

*The nudging relaxation time scales for ECHAM6 in the standard setup used here are 6 h for vorticity, 48 h for divergence, and 24 h for surface pressure. Here the model is not retuned with nudging. However, regional dust correction factors are different for the NUDGED and CLIM experiments (this was added in the text). For this study it was considered to use the nudged model version to test the aerosol module with a transport that is as realistic as possible. For any studies investigating interactions of aerosol and climate the non-nudged 'climate' version will be used.*

Page 13, Line 13-16, The 2 nudged simulations produce very large AOD bias over subtropical ocean (Fig. 3). This might indicate that the sea salt emission needs to be tuned down.

*Too high sea salt emissions in this region may be a problem causing too high AOT in this region. The discussion of the model performance of sea salt aerosol was extended in the Results section. Other causes for overestimating AOT in this region may originate from too high aerosol hygroscopic growth*

*(as the model does not use a limitation for very high relative humidities) or low aerosol removal by wet deposition, which would have a noticeable effect in this region. This discussion has been expanded in the text.*

Section 5.3: I think near the source region the size distribution comparison is not very interesting since the model's size distribution is closely related to the assumed size distribution at emission. The comparison between more interesting when it is done for remote regions when the model physics has enough time to change the size distribution. I wonder if it is possible to do the analysis that way. Could you please comment?

*The EUSAAR sizes are generally close to source regions. In this respect the comparison of the simulated aerosol size distributions to the data compilation for marine regions by Heintzenberg et al (2000) represents the aerosol size evolution at remote regions, as suggested by the reviewer. It is notable that the marine measurements generally show more defined and separated Aitken and accumulation modes. This can be attributed to aerosol being 'aged' at these remote regions. The pronounced differences between the modes is reproduced in the model results in most regions. The size distribution for the Aitken mode is wider in the model than in the observations, which points to an overestimate of the width of the Aitken mode in the model by the prescribed mode standard deviation of 1.59.*
*Compared to the previous version of ECHAM-HAM described in Zhang et al (2012) who evaluated the model results with the same dataset in their figure 16, the simulated size distribution of the accumulation mode now better matches the observations for the latitude band 45-60S, which is an indicator for an improvement of marine aerosol in the new model version (see also Figure 25).*

*The section 5.3 describing the evaluation of marine aerosol size distribution is rewritten/expanded accordingly to make these points more clearer*

Figure 5, 6, 19, 20, 22a,b: Please also add the R, RMSE, and mean bias in the figure.

*Done*

**Reviewer 3 (Ron Miller):**

This is a well-written article that represents an impressive amount of work to collect a wide range of observations for evaluation of an aerosol model. I have some suggestions for improving clarity, but the article should be suitable for publication subject to minor revision. If the authors have any questions, they can contact me at ron.l.miller@nasa.gov.

1. The article is a detailed and extensive evaluation of the current model. The article could have even more lasting value if it anticipated and aided development of future model versions.

My main suggestion is to add more quantitative metrics to certain figures. These metrics can be useful when future generations of the aerosol model are developed and evaluated. For example, Figures 19 and 20 will be of limited use in future development, because the current model assessment is mainly visual and qualitative (i.e. are the points clustered around the 1:1 line?). These figures would benefit from the summary statistics that are listed in Figures 21 or 23 (e.g.), so that similar figures of future model versions can be compared using these statistics. This suggestion also applies to Figures 5, 6 along with 22a and b. Figures 4, 8, 18 and 24 could also be improved with summary metrics. More generally, the authors should think about how they might assess future model versions and augment the figures so that any eventual improvement can be quantified. The article would also benefit from a discussion in the conclusions about what the

authors consider to be key model errors that should guide future development.

*We thank Dr. Miller for the thorough assessment and the constructive comments. We followed almost all of the technical suggestions, as detailed below. Statistical information was added to figures 5,6,19,20 and 22, but was omitted for the time series plots.*

2, 18: add citation to Huneeus et al. ACP 2011?

*Done*

2, 30: somewhere in this paragraph, it would be helpful to state explicitly that the fields calculated by the chemistry module MOZ are prescribed in this study, so that only the aerosol calculation is fully interactive. (I suggest this distinction because Section 2 describes the capabilities of both the HAM and MOZ modules: a joint description that I like for placing the HAM module in context.)

*We added 'In this study only the aerosol module HAM is used, such that the aerosol computations are fully interactive, while the oxidant fields that would be computed interactively in the HAMMOZ setup are prescribed in this setup' to the paragraph.*

3, 1: add 'or bin' after 'sectional'?

*Done*

Section 2.2 ('ECHAM'): it would be helpful if the horizontal and vertical resolution of the model version used in this study was listed here, rather than postponed to Section 3.

*Following the suggestion, we added this information to Section 2.2, but also retained it in Section 3.*

Table 1 caption: 'Mode boundaries for the ' missing word?

*Replaced sentence with 'Mode boundaries for the number median particle radii R are given for each mode'.*

5, 20: 'parameterization for *ocean* temperature dependence'?

*Done*

6, 8: 'individual sectors'? Define or give examples of 'sectors' here (instead of in the following paragraph)?

*Done*

6, 21: 'T63' This is one reason why the ECHAM resolution would be more helpfully defined in Section 2.2 than Section 3.

*The model resolution is now already defined in Section 2.2*

6, 27: 'Interannual variability of biomass burning is not considered.' But is the decadal averaged burning updated each year via interpolation so that it doesn't change abruptly each decade?

*Yes. The explanation 'Interannual variability of biomass burning is not considered, but the decadal emissions are interpolated for the individual years' was added*

7, 13: 'Dust particle emissions are driven by...' How is the ratio of emitted silt particle sizes to clay sizes determined? This is relevant to the coarse-mode burden. Which later is found to be unrealistically small.

*Usually dust particles smaller than 1 μm radius are considered as clay-sized and larger dust particles are silt-sized particles. The size mode limit in HAM between accumulation and coarse mode particles is 0.5 μm radius. Thus there is no direct correspondence between the aerosol modes in the model and the clay and silt dust particle sizes. The emitted size distribution between dust in the different modes is determined by the emission module and takes into account surface wind speeds and threshold velocities that depend on soil conditions. However, only a very small part (about 1%) of the simulated dust emission is emitted in the accumulation mode (see Stier et al., 2005). The underestimated dust particle size are likely caused by an underestimation of the coarse mode particle radius.*

7, 19: 'Tegen et al. (2002), who identified potential source areas...' It's worth noting that the sources are also identified using a calculation of paleo-lake extent (that I consider innovative).

*Do*ne

7, 30: 'These regional correction factors...' What observations and criteria were used to arrive at this correction?

*The regional correction factors were chosen such that the emissions in the specific regions agree with the emissions by Huneeus et al. (2011), in which the orography was taken into account to compute the surface roughness, which is an important factor in computing dust emissions. The explanation was modified in the text to make this clearer. The underlying problem that requires regional correction factors for the dust emission is the change in the surface parameterization as a consequence of coupling the atmospheric model ECHAM with the land surface scheme JSBACH. In the version ECHAM6.3 the roughness length definition was updated. This setting was adapted to ECHAM6.3-HAM2.3. With the changed roughness parameterization different regions became active dust sources compared to the previous version, as the threshold velocities for dust and surface wind shear patterns were modified.*

8, 5: 'As a marine source...' What physical variables control emission of DMS? (wind speed? ocean temperature?)

*The DMS emission depends on seawater DMS concentration, and 10-m wind speed. Air-sea exchange of DMS is parameterized according to Nightingale et al (2000), GBC. This part was expanded in the text.*

Section 2.3.2 ('Aerosol Microphysics') The omission of nitrate aerosols should be noted, since this could contribute to an underestimate of PM or AOT.

*We added the sentence 'Nitrate that may also form secondary ammonium nitrate aerosol is currently not considered in HAM' to the section.*

9, 11: 'in-cloud scavenging scheme' Observed scavenging depends upon whether the aerosol is hydrophilic, and this will vary with aerosol species (with sulfates being very hydrophilic and some common dust minerals as hydrophobic). Does droplet activation account for varying aerosol composition within each size mode?

*The in-cloud scavenging scheme described in Croft et al (2010) distinguishes between the aerosol modes considering the particle size and whether the aerosol mode is 'soluble' or 'insoluble'. Thus, the change from hydrophobic to hydrophilic particles by condensation processes is considered. The activation scheme also accounts for the aerosol composition within the aerosol modes.*

9, 20: '1.52 + 0.0011' It should be noted that this represents relatively little shortwave absorption by dust, in better agreement w/ AERONET retrievals (Sinyuk et al GRL 2003) than the original Patterson et al. (1977) laboratory measurements of far-traveled Saharan dust samples.

*This is true, and partly caused by the fact that the parameterization of aerosol radiative properties in the model described by Kinne et al (2013) utilized the results by Sinyuk et al (2003) for dust. A note on this is added to the text. However, if the dust particle size is underestimated in the model the resulting dust single scattering albedo may actually turn out to be too low in the model.*

9, 27: Does dust nucleate ice particles by enabling heterogeneous freezing? Does nucleation vary with the aerosol mixture in each size mode, given that some aerosols like dust are much more efficient?

*In ECHAM6.3-HAM2.3 contact ice nucleation can be triggered only by mineral dust, and only dust and black carbon particles can act as immersion ice nuclei at mixed-phase cloud temperatures (0° C to -35° C). A detailed description of the description of ice nucleation processes in ECHAM6-HAM2 is in the publication by Lohmann and Neubauer, 2018, ACP. Here an explanatory sentence was added.*

10, 14: 'are relaxed' What is the time scale for relaxation? Does it vary with height?

*The nudging relaxation time scales for ECHAM6 in the standard setup used here are 6 h for vorticity, 48 h for divergence, and 24 h for surface pressure; relaxation times do not vary with height.*

10, 21: 'do not vary on daily or interannual time scales' But is emission updated each year to reflect slow decadal trends, or is emission held constant over the simulation period?

*Yes, correct. As already done in the methods section, this aspect was again added here.*

10, 22: After 'satellite retrievals' insert '(labeled "GFAS")'?

*Done*

11, 9: 'respective AERONET stations' How many years of measurements are available for the stations? Do some stations have short records that may not be representative of the simulation decade?

*Some stations have short records. However the measurements and model results are compared for the same time periods also in the scatter plots, so that it should not lead to an overall bias in the comparisons.*

11, 20: 'and with compiled number size distributions...by Heintzenberg et al. (2000)' Maybe move this phrase down as few sentences to where this measurement set is discussed in detail?

*We prefer to leave the reference to the size distributions by Heintzenberg et al (2000) at the top of the paragraph to provide a brief introduction that to the two datasets that are used to compare the aerosol size distributions.*

12, 7: IMPROVE. it should be noted that some of these sites are at elevation in regions where the topography is not well-resolved by the model.

*Done*

Also, why did you not evaluate dust using these measurements? e.g. see:
VanCuren, R., and T. Cahill, Asian aerosols in North America: Frequency and concentration
of fine dust, J. Geophys. Res., 107(D24), 4804, doi:10.1029/2002JD002204,
2002.

*Thank you for the reference. Here we did not use the dust data from the North American stations for evaluation as it can be expected that these would be mainly influenced by local dust sources which are anthropogenically influenced, which is not taken into consideration in this model setup. However we agree that it may be an interesting future study to assess simulations of long-range Asian dust events recorded in North American measurements.*

13, 5: Why restrict the comparison of MODIS AOT to 2007? Is this year representative
of the entire simulation?

*To limit the extent of the paper only one year was chosen for the comparisons. 2007 is a quite typical year for dust emissions.*

13, 17: 'This may point to missing aerosol sources' Maybe say 'aerosol species' instead
of aerosol 'sources'? Also, Bauer et al. (2015) argue that nitrate aerosols and
ammonium sulfate represent a significant fraction (a little more than half) of the anthropogenic
contribution to PM2.5 over central North America. The omission of these
aerosol species from HAM could contribute to the underestimate of AOT.
Bauer, S. E., K. Tsigaridis, and R. Miller (2016), Significant atmospheric aerosol
pollution caused by world food cultivation, Geophys. Res. Lett., 43, 5394–5400,
doi:10.1002/2016GL068354.

*Agreed, 'sources' was changed to 'species' in the text. Missing consideration of ammonium nitrate is indeed a problem that may lead to underestimate the AOT. We emphasize that nitrate aerosol should be added to HAM as priority for future model development. Also thank you for the reference, which was added to the manuscript.*

Figure 4 caption: 'Error bars show the variabilities of the measurements.' How is the
variability defined? Standard deviation of daily values?

*Yes, this was added to the figure caption.*

14, 16: 'slightly'? 0.01 seems like a large improvement compared to -0.3.

*True. We removed 'slightly'.*

Figure 8: Could labels be added to the top of each column explaining the quantity
plotted (AOT, AE and SSA)?

*Done*

15, 16: 'underestimating the particle size' Is it known why this discrepancy over the
Sahara is largest in the NH Fall?

*The underestimation of aerosol particle size in the NH fall season may be related to too low dust*

*emissions in the Sahara, which is reflected by too low AOD compared to the observations. Thus the AE signal is dominated by transported (relatively small) anthropogenic aerosol such as sulfate from anthropogenic fossil fuel or wood burning. Too low dust emissions in this season in turn may be related to anunderestimation of emission events caused by moist convection, which cannot be well represented by the parameterized convection in the model. We added this explanation to the text.*

15, 22: 'the too low particle size ... could result in too high SSA' Why would coarse
particles reduce the SSA? Is there a simple explanation or reference for this?

*Following results from Mie calculations e.g, shown in Lacis and Mishchenko (1995) supermicron dust particles are more absorbing and have thus lower SSA compared to submicron dust particles for particles with the same complex refractive indices. This explanation and the reference were added in the text.*

15, 34: 'too strong mixing' vertical or horizontal or both?

*In this region a misrepresentation of both horizontal and vertical mixing can be expected if the topography is not highly resolved.*

Figure 10 caption: replace 'for over' with 'over'?

*Done*

16, 6: 'may not be representative' This is especially true for stations where topography
is poorly resolved.

*Yes, this was already mentioned above.*

16, 26: Table 3 includes median values from AeroCom. It should be stated that this
median is derived from models and is not necessarily in agreement with observations.
Also, this comparison between HAM2.3 and AeroCom would be more useful is some
measure of model diversity was added to the AeroCom column of Table 3.

*We added the standard deviation as measure for the range of AeroCom results to the table.*

16, 30: 'However...emission fluxes also depend on the size range considered...' This is
a very relevant caveat. To me, it suggests that the comparison of HAM and AeroCom emission be
replaced by a comparison of aerosol burden.

*The aerosol burdens are also compared in this table. While it is true that the usefulness of comparisons of dust emission fluxes for diverse particle size ranges is limited, we nevertheless want to keep the emission comparison, as these numbers are often cited. Overall for dust emission comparisons it would make sense to always compare the emissions for individual size ranges, but this information is not available for all models.*

16, 34: 'in agreement with the AeroCom average burden of 19.2 Tg' Again, the AeroCom
burden is not necessarily indicative of the observed value. As an alternative,
Kok et al. 2017 calculate a global burden of around 20 Tg that is better constrained by
observations: Kok, J.F., D.A. Ridley, Q. Zhou, R.L. Miller, C. Zhao, C.L. Heald, D.S. Ward, S.
Albani, and K. Haustein, 2017: Smaller desert dust cooling effect estimated from analysis of

dust size and abundance. Nature Geosci., 10, no. 4, 274-278, doi:10.1038/ngeo2912.

*We added this reference as it contains an updated constraint based on measurements compared to the published AeroCom values. However considering the AeroCom median, it is encouraging that the new constrained value of 20 Tg dust burden is actually quite close to the AerocCom average burden of 19 Tg, given the ranges of model diversity for dust burdens.*

17, 14: 'R on log scale' Does this mean that R is calculated by correlating logarithms of concentration and not concentration itself? This is fine, but it should be explained that such a correlation emphasizes the ability of a model to correctly simulate variations in concentration over large distances from the source, rather than subtle differences over more limited regions.

*Done*

Figure 12 caption: 'Simulated vertical profiles are shown as monthly and regional averages.'
What is the duration of the flight data sets that are compared to the monthly
averaged model concentration in this figure?

*The durations of flight campaigns vary but are usually for a few weeks (for details see Heald et al 2011), thus the timescales of the flight data and the monthly model values are comparable.*

Table 3 caption: does 'sedimentation flux' refer to gravitational settling?

*Yes, changed in the table caption*

19, 8: 'while the CLIM and NUDGE results underestimate BC' The values from these
simulations are indeed smaller than the GFAS values, but all seem to be with in the
uncertainty of the measurements.

*True. We changed the formulation in the text accordingly.*

20, 15: 'the correlation is better' ... 'R values of 0.78 for the correlation of annual AEs for both the
CLIM and NUDGE simulation' If the correlations are the same, how can one be better?

*True, this is removed.*

21, 6: 'my be' (may be?)

*corrected*

Why 'may be'? Doesn't Soffiev attribute the temperature dependence to specific physical processes
like surface tension and solubility (or is the temperature dependence entirely empirical)?

*The temperature dependence observed by Sofiev (and others) so far is just empirical, derived from laboratory experiments. No definitive explanation has been given by the authors.*

21, 18: 'spume drops' Could you define spume drops and their relation to the emitted
size distribution for the benefit of readers like myself :) who are not specialists in seasalt
aerosols?

*We followed the suggestion and added some explanation of spume drops to the text. 'Spume drops are teared off wave crests at high wind speeds, thus their emission is related to wave breaking. These spume droplets have particle sizes on the order of 20 µm or larger (see e.g. Andreas et al 2010). However, they are not expected to be relevant for the atmospheric aerosol burden because their large sizes cause them to sediment quickly resulting in a very short lifetime. Also their impact on both radiative fluxes and as CCN is expected to be small. Nevertheless, including spume drop formation in the sea salt emission parameterization may lead to high emission mass fluxes, while sea salt aerosol number concentrations are not strongly affected by the spume drop formation. The omission of spume drops in the new sea salt emission parameterization may explain much lower sea salt emission fluxes in this model version compared to earlier versions.'*

page 22: the years of comparison for sea salt are confusing. Line 1 cites the year
2010. But line 5 claims that 2006 is used in Figure 22, whose caption says 2007. In
fact, Figure 23 does seem to be based on 2010 according to the caption, but this is not
noted in the text (line 15 onward).

*Thanks for pointing this out. The year 2006 was wrong, the comparisons was done for 2007. This was corrected. The reason for using 2007 for the comparison was that more data were available for the marine aerosol network. The results for 2006 are similar, but have less data points. The reference to year 2010 for comparing the surface concentration data the Figure 23 was added in the text.*

22, 22: 'For high latitude stations ...' This generalization doesn't seem to be true for
the only NH high latitude station. Also, why are the NUDGE and Gong-T models with
better physics behaving so poorly at Cape Grimm (41 s), where temperature effects
might be expected to be important? The improved agreement at Marion Island, just 6
degrees poleward, suggests individual stations might include large regional effects that
are unique to each island.

*Indeed the measurements of sea salt concentrations may be strongly influenced by local effects that influence the particle transport to some of the island stations. It should however still be noted that the performance at other stations at high latitudes is consistently better for the new sea salt emission parameterization compared to the previous version, such that this is an indicator for the necessity of including a temperature-dependent sea salt emission scheme.*

23, 21: 'Mineral dust and sea salt aerosol distributions must be well characterized in
order to be able to make meaningful statements on the importance of anthropogenic
aerosol effects.' I agree with this, but could you justify this point more fully? For climate
change, we don't need to characterize natural sources if their radiative forcing is time-independent. To
be sure, dust sources expand and contract with time (e.g. due to variations in hydroclimate).
Nonetheless, it is hard to evaluate simulated anthropogenic species if the natural species are poorly
simulated, because common observed variables like AOT or PM2.5 include the effect of all species
together. (Thus, most AeroCom models get reasonable values of AOT despite widely varying fractions
of natural and anthropogenic species.)

*We added: 'In addition to the need for understanding the distribution of natural aerosols in order to evaluate anthropogenic aerosol distributions, also anthropogenic aerosol effects such as aerosol-cloud interactions depend not only on the anthropogenic enhancement of aerosols, but also on the background aerosol from natural sources. Natural aerosol emissions of dust, but also sea salt or vegetation emissions may change in a changing climate due to changing wind patterns or surface conditions. A realistic representation of the processes controlling the emissions and atmospheric distribution of the natural aerosols is needed as a basis for reliable projections of aerosol-climate interactions in a changing climate.'*